# NLRP3 phosphorylation in its LRR domain critically regulates inflammasome assembly

Tingting Niu [1,2], Charlotte De Rosny [1], Séverine Chautard [1], Amaury Rey [1], Danish Patoli[1], Marine Groslambert[1], Camille Cosson [1], Brice Lagrange[1], Zhirong Zhang[3], Orane Visvikis[4], Sabine Hacot[5], Maggy Hologne [6], Olivier Walker [6], Jeimin Wong [2], Ping Wang[7], Roméo Ricci[3], Thomas Henry [1], Laurent Boyer [4], Virginie Petrilli [5] & Bénédicte F. Py [1✉]

NLRP3 controls the secretion of inflammatory cytokines IL-1β/18 and pyroptosis by assembling the inflammasome. Upon coordinated priming and activation stimuli, NLRP3 recruits NEK7 within hetero-oligomers that nucleate ASC and caspase-1 filaments, but the apical molecular mechanisms underlying inflammasome assembly remain elusive. Here we show that NEK7 recruitment to NLRP3 is controlled by the phosphorylation status of NLRP3 S803 located within the interaction surface, in which NLRP3 S803 is phosphorylated upon priming and later dephosphorylated upon activation. Phosphomimetic substitutions of S803 abolish NEK7 recruitment and inflammasome activity in macrophages in vitro and in vivo. In addition, NLRP3-NEK7 binding is also essential for NLRP3 deubiquitination by BRCC3 and subsequently inflammasome assembly, with NLRP3 phosphomimetic mutants showing enhanced ubiquitination and degradation than wildtype NLRP3. Finally, we identify CSNK1A1 as the kinase targeting NLRP3 S803. Our findings thus reveal NLRP3 S803 phosphorylation status as a druggable apical molecular mechanism controlling inflammasome assembly.

[1] CIRI, Centre International de Recherche en Infectiologie, Univ Lyon, Inserm, U1111, Université Claude Bernard Lyon 1, CNRS, UMR5308, ENS de Lyon, F-69007 Lyon, France. [2] Shanghai Key Laboratory of Regulatory Biology, Institute of Biomedical Sciences and School of Life Sciences, East China Normal University, 500 Dongchuan Road, 200241 Shanghai, China. [3] IGBMC, Institut de Génétique et de Biologie Moléculaire et Cellulaire, CNRS, UMR7104, Inserm, U964, Université de Strasbourg, Illkirch, France. [4] Université Côte d'Azur, Inserm, C3M, F-06204 Nice, France. [5] CRCL, Centre de Recherche en Cancérologie de Lyon, INSERM U1052, CNRS UMR5286, Université de Lyon, Université Lyon 1, Centre Léon Bérard, Lyon, France. [6] Institut des Sciences Analytiques (ISA), Univ Lyon, CNRS, CNRS UMR5280, Université Claude Bernard Lyon 1, Villeurbanne, France. [7] Shanghai Tenth People's Hospital of Tongji University, Tongji Cancer Center, School of Medicine, Tongji University, 200092 Shanghai, China. ✉email: benedicte.py@inserm.fr

Innate immunity constitutes a highly efficient barrier to diverse insults by rapidly detecting pathogens as well as tissue damage. The activation of pattern recognition receptors (PRRs) by pathogen- and damage-associated molecular patterns (PAMPs and DAMPs) leads to the early local detection of the insult and the production of pro-inflammatory cytokines. NLRP3 is a protective PRR against infections by highly diverse agents including viruses, bacteria, fungi and parasites. On the other side, NLRP3 activity is detrimental in several inflammatory conditions. Indeed, gain-of-function mutations in the *NLRP3* gene cause Cryopyrin-associated periodic syndrome (CAPS) autoinflammation, and inappropriate activation of NLRP3 contributes to a number of multifactorial diseases including gouty arthritis, atherosclerosis, diabetes, neurodegenerative disorders, and ischemia-reperfusion insults[1]. Upon activation, NLRP3 assembles a typical multimeric inflammasome complex comprising NEK7, the adapter ASC and the effector caspase-1. Caspase-1 then controls the cytosolic maturation of the pro-inflammatory cytokines IL-1β and IL-18, and processing of gasdermin D (GSDMD). Cleavage of GSDMD releases its N-terminal domain forming pores into the plasma membrane that mediate the release of mature cytokines and trigger pyroptosis leading to alarmin release.

The regulation of NLRP3 inflammasome assembly remains incompletely understood. The variety of conditions leading to NLRP3-dependent inflammation and the structural diversity of its known activators suggest that NLRP3 senses a common cell stress downstream of these conditions. NLRP3 is both regulated at the transcriptional and post-translational levels[2,3]. In particular, NLRP3 activation requires the deubiquitination of its LRR domain by the JAMM domain-containing $Zn^{2+}$ metalloprotease deubiquitinase (DUB) BRCC3[4]. Since this study, accumulative studies have pointed out the role of post-translational modifications of NLRP3 in its activation process, including modifications of the NLRP3 LRR domain. Indeed, in addition to BRCC3-mediated NLRP3 LRR deubiquitination, which is required for the inflammasome assembly, NLRP3 LRR has been shown to be ubiquitinated and targeted to degradation by FBXL2 at K689 (in mouse NLRP3 corresponding to K691 in human NLRP3) as well as by MARCH7 and RNF125. Reversely, ubiquitination of NLRP3 LRR domain by Pellino2 participates in NLRP3 priming[5–8]. In addition to ubiquitination, the phosphorylation status of NLRP3 LRR domain regulates the inflammasome as PTPN22-mediated dephosphorylation at Y859 (in mouse NLRP3 corresponding to Y861 in human NLRP3) promotes its activation[9]. Most of the modified sites remain to be identified.

In this study, we analyzed the post-translational modifications of the NLRP3 LRR domain by mass spectrometry and identified serine 803 (S803) in mouse (S806 in human NLRP3) to be phosphorylated upon priming and dephosphorylated upon activation. S803 was critical for NLRP3 activation in human monocytes and mouse macrophages using reconstitution of NLRP3-deficient lines. Consistently, primary macrophages from $Nlrp3^{S803D/S803D}$ knock-in (KI) mice bearing a phospho-mimetic substitution of S803 neither underwent pyroptosis nor secreted IL-1β and IL-18 in response to various inflammasome activators. S803D substitution impaired inflammasome assembly and caspase-1 processing and activation. $Nlrp3^{S803D/S803D}$ KI mice were resistant to in vivo endotoxic shock. Further analysis of the mechanism revealed that, NLRP3 S803D failed to recruit NEK7 which constituted a prerequisite for BRCC3 recruitment upon activation. NLRP3 S803D was then found highly ubiquitinated and targeted to degradation. Lastly, combining bioinformatic analysis, siRNA screen and biochemistry assays, we identified the ubiquitous CSNK1A1 kinase as critical for NLRP3 phosphorylation at S803. Altogether, our results revealed phosphorylation at S803 as a critical upstream regulatory checkpoint in NLRP3 activation process.

## Results

**NLRP3 LRR is phosphorylated at S735, S806, and S1035.** We used mass spectrometry in order to identify novel modification sites in the LRR domain of NLRP3 that could be linked to ubiquitination. 293T cells expressing Flag-NLRP3 LRR were treated with deubiquitinase inhibitor G5, proteasome inhibitor MG132 and lysosomal protease inhibitor E-64d to increase NLRP3 LRR ubiquitination level. Flag-NLRP3 LRR purified by anti-Flag immunoprecipitation (IP) was analyzed by SDS-PAGE followed by Western-blot (WB) (Supplementary Fig. 1a) and Coomassie staining (Fig. 1). Proteins in the gel sections containing Flag-NLRP3 LRR or the smear of ubiquitinated Flag-NLRP3 LRR were digested by trypsin, separated by reverse-phase ultra-high performance liquid chromatography and analyzed by mass spectrometry (LC-MS/MS). We identified three ubiquitination sites K878, K927, and K973 (in human, corresponding to K875, T924, and K970 in mouse respectively) and three phosphorylation sites S735, S806, and S1035 (in human, corresponding to S733, S803, and S1032 in mouse respectively) (Table 1 and Supplementary Fig. 1b). Two of these ubiquitination sites and all identified phosphorylation sites were conserved across species and exposed at the surface of the LRR domain (Supplementary Fig. 1c, d).

**S806 is critical for NLRP3 inflammasome activation in human monocytes.** In order to test the impact of these modifications on NLRP3 inflammasome activation, we used U937 human monocytes knocked-out for *NLRP3* by Crispr/CAS9 and reconstituted them with doxycycline-inducible NLRP3 bearing substitutions of these sites (Supplementary Fig. 2a)[10]. K878R, K927R, or K973R substitutions of the three identified ubiquitinated lysines by arginines did not alter significantly IL-1β secretion in LPS-primed U937 treated with nigericin. These results suggested that ubiquitination of these lysines were not critical for NLRP3 inflammasome activation. In contrast, the analysis of the three phosphorylated serine sites showed that NLRP3 phospho-

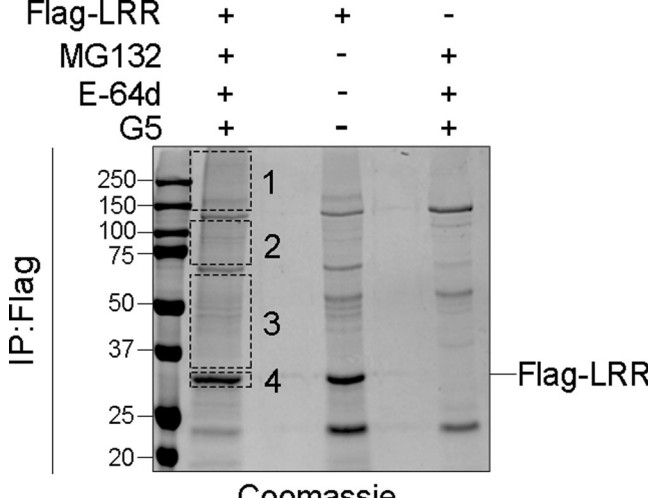

**Fig. 1 NLRP3 is ubiquitinated at K878, K927, K973, and phosphorylated at S735, S806, S1035.** 293T cells were transfected with plasmids coding for Flag-LRR. One day later, cells were treated with MG132 (10 μM), E-64d (20 μg/ml), and G5 (1 μM) for 30 min. Flag-LRR was purified by anti-Flag immunoprecipitation and analyzed by SDS-PAGE stained with Coomassie solution. Smear corresponding to ubiquitinated Flag-LRR (sections 1–4, excluding major unspecific bands) was analyzed by mass spectrometry. Data correspond to the one selected experiment used to perform the mass spectrometry analysis, out of 8 independent repeats. Molecular weights are indicated in kDa. IP:Flag anti-Flag immunoprecipitates.

**Table 1 Identification of three ubiquitination sites and three phosphorylation sites by LC-MS/MS analysis by mass spectrometry.**

| Section | Peptide sequence | Modification | Site |
|---|---|---|---|
| 1/ above 115 kDa | R.LYVGENALGDSGVAILCEK@AK.N | Ubiquitination | K 878 |
| | R.K@LSLGNNDLGDLGVM*M*FCEVLK.Q | Ubiquitination | K 973 |
| | R.GNTLGDK@GIK.L | Ubiquitination | K 927 |
| 2/ 70–115 kDa | R.LYVGENALGDSGVAILCEK@AK.N | Ubiquitination | K 878 |
| | R.GNTLGDK@GIK.L | Ubiquitination | K 927 |
| 3/ 30–70 kDa | R.GNTLGDK@GIK.L | Ubiquitination | K 927 |
| 4/ 30 kDa | R.GNTLGDK@GIK.L | Ubiquitination | K 927 |
| | R.GLFS#VLSTSQSLTELDLSDNSLGDPGM*R.V | Phosphorylation | S 735 |
| | K.LVELDLS#DNALGDFGIR.L | Phosphorylation | S 806 |
| | K.SALETLQEEKPELTVVFEPS#W.- | Phosphorylation | S 1035 |

mimetic S806D fully abrogated IL-1β secretion in LPS-primed U937 treated with nigericin, while phospho-null or phospho-mimetic substitutions of S735 and S1035 showed no or poorly reproducible impacts on IL-1β secretion (Fig. 2a). As expected, all reconstituted U937 lines and wild-type (WT) U937 secreted equal amounts of TNF used as a control for NLRP3-independent signaling in response to LPS priming. Noteworthy, NLRP3 activity depends on NLRP3 LRR domain in this U937 reconstitution system matching endogenous NLRP3 expression level (Supplementary Fig. 2b). Follow-up on S806 substitutions confirmed that phospho-mimetic S806D and S806E mutants impaired mature IL-1β and IL-18 secretion and release of cleaved caspase-1 in LPS-primed U937 treated with nigericin, while phospho-null S806A mutant inhibited but did not abolish these responses (Fig. 2b, c). In addition, ASC speck formation was also impaired in LPS-primed U937 expressing NLRP3 S806D treated with nigericin (Fig. 2d). S806 substitutions impaired IL-1β and IL-18 secretion by U937 irrespectively of the LPS priming pathway used prior to nigericin treatment. Indeed, U937 expressing S806A, D, or E mutants primed with 10 min LPS treatment (non-transcriptional MyD88-dependent priming), with 30 min or 1 h LPS treatment (non-transcriptional TRIF-dependent priming), with 3 h LPS treatment (transcriptional priming) were impaired in IL-1β and IL-18 secretion (Supplementary Fig. 2c)[11]. In addition, S806 substitutions impaired IL-1β and IL-18 secretion by U937 irrespectively of the nature of the priming signal as evidenced by priming cells with TLR2/6 ligand Pam3CSK4 prior to nigericin treatment (Supplementary Fig. 2d). S806 substitutions also impaired IL-1β and IL-18 secretion in LPS-primed U937 cells treated with particulates including monosodium urate (MSU) and silica (Supplementary Fig. 2e). On the opposite, LPS-primed U937 expressing NLRP3 S806A, D, or E mutants secreted IL-1β as WT U937 in response to infection by *Salmonella enterica serovar Typhimurium* (SL1344 strain) and transfection with poly(dA:dT), activating the NLRC4 and AIM2 inflammasomes respectively (Supplementary Fig. 2f, g). Altogether, these data showed that NLRP3 S806 phospho-null and phospho-mimetic mutants strongly impaired the NLRP3 inflammasome activation. We then monitored NLRP3 phosphorylation upon priming and activation of the inflammasome in reconstituted U937. While NLRP3 was not phosphorylated in basal condition, it got phosphorylated on serine upon priming with LPS, and further activation with nigericin strongly reduced this phosphorylation (Fig. 2e). No phosphorylation was detected in U937 reconstituted with NLRP3 S806A, indicating that S806 is critical in this priming-dependent phosphorylation. Consistently with this result, we did not detect the reported phosphorylation of NLRP3 S198 upon LPS priming (Supplementary Fig. 2h, i)[12].

**S803 is critical for NLRP3 inflammasome activation in immortalized BMDMs.** In parallel, we also tested the impact of these modifications in murine bone marrow-derived macrophages (BMDMs) by reconstituting immortalized BMDMs from *Nlrp3*^−/− mice with doxycycline-inducible mutants of murine NLRP3 (Supplementary Fig. 3a). Consistently with results obtained in U937 human cell lines, immortalized BMDMs expressing NLRP3 with substitutions of S733 or S1032 secreted IL-1β in response to LPS priming followed by activation with nigericin similarly as BMDMs expressing WT NLRP3 (Fig. 3a). In contrast, S803A, S803D, and S803E substitutions fully abolished IL-1β secretion in LPS-primed immortalized BMDMs treated with nigericin or extracellular ATP confirming that S803 is critical for NLRP3 activity (Fig. 3a, b). Noteworthy, NLRP3 S803A mutant showed no residual activity in mouse immortalized BMDMs as opposite to the corresponding NLRP3 S806A substitution in human U937 line, suggesting some species-dependent pathway. Similarly as in U937, NLRP3 activity depends on NLRP3 LRR domain in this BMDM reconstitution system matching endogenous NLRP3 expression level (Supplementary Fig. 3b). As expected, LPS-primed immortalized BMDMs expressing S803A, S803D or S803E mutants secreted equal amounts of IL-1β as compared to BMDMs expressing WT NLRP3 upon *S. Typhimurium* SL1344 infection or transfection with flagellin activating the NLRC4 inflammasome, as well as upon transfection with poly(dA:dT) activating the AIM2 inflammasome (Supplementary Fig. 3b–d).

**NLRP3 S803D mutant is defective for inflammasome assembly in primary BMDMs.** We generated *Nlrp3*^S803D/S803D KI mice by Crispr/CAS9 (Supplementary Fig. 4a, b). Primary BMDMs from *Nlrp3*^S803D/S803D mice were totally impaired in IL-1β and IL-18 secretion when primed with LPS and activated by ATP, nigericin, LLOMe or MSU. They did not undergo pyroptosis upon LPS and nigericin treatment (Fig. 4a, b). BMDMs from *Nlrp3*^S803D/+ mice showed an intermediate phenotype. As a control, LPS-induced TNF secretion was identical in all BMDMs. LPS-primed *Nlrp3*^S803D/+ BMDMs treated with nigericin secreted similar amount of IL-1β and IL-18 than *Nlrp3*^+/− BMDMs, confirming the loss-of-function of the *Nlrp3*^S803D allele (Supplementary Fig. 4c). *Nlrp3*^S803D/S803D BMDM phenotype was independent of the nature of the priming signal, as similar results were obtained when Pam3CSK4-primed BMDMs were treated with nigericin (Supplementary Fig. 4d). Consistently, no mature IL-1β and cleaved caspase-1 could be observed by WB in the supernatants of *Nlrp3*^S803D/S803D BMDMs primed with LPS and activated with nigericin (Fig. 4c). In addition, nigericin did not trigger caspase-1 activity in LPS-primed *Nlrp3*^S803D/S803D BMDMs as monitored by staining with FLICA fluorescent caspase-1 substrate (Fig. 4d). We next tested whether *Nlrp3*^S803D/S803D BMDMs were impaired in the assembly of the NLRP3 inflammasome, and indeed ASC did not form oligomers in LPS-primed *Nlrp3*^S803D/S803D BMDMs treated with nigericin, and no ASC speck could be observed

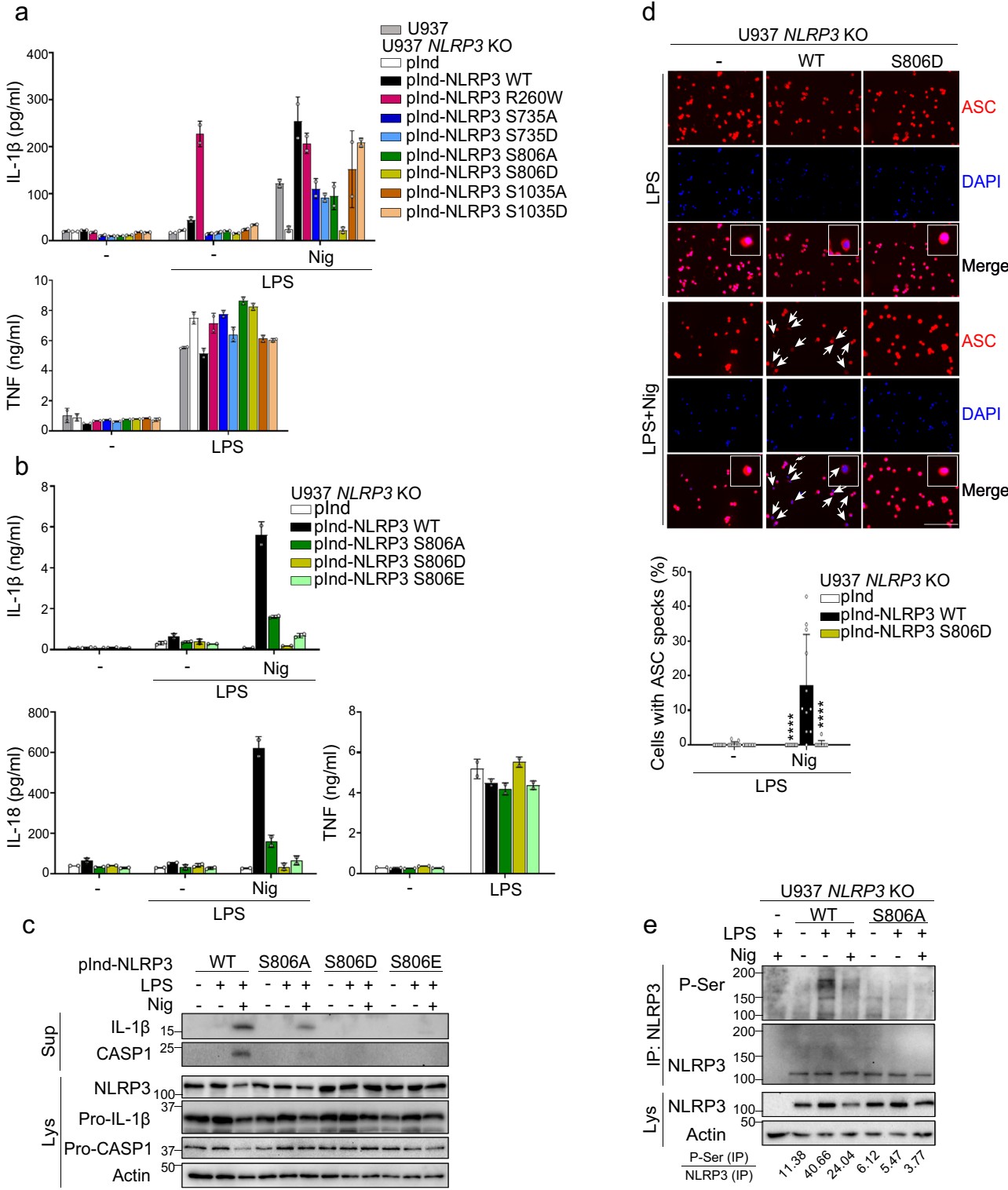

(Fig. 4e, f). In addition, NLRP3 S803D did not co-immunoprecipitate with ASC in $Nlrp3^{S803D/S803D}$ BMDMs treated with LPS and nigericin (Fig. 4g). Consistently, LPS-primed $Nlrp3^{S803D/S803D}$ BMDMs infected with *Escherichia coli*, or Pam3CSK4-primed $Nlrp3^{S803D/S803D}$ BMDMs transfected with LPS, did not secrete IL-1β which relies on the NLRP3 inflammasome, but underwent pyroptosis which depends in both settings on the caspase-11 non-canonical inflammasome (Supplementary Fig. 4e, f). As controls, LPS-primed $Nlrp3^{S803D/S803D}$ BMDMs secreted IL-1β in response to *S. Typhimurium* infection or flagellin

transfection both activating the NLRC4 inflammasome (Supplementary Fig. 4g, h). In addition, LPS-primed $Nlrp3^{S803D/S803D}$ BMDMs secreted IL-1β and underwent pyroptosis in response to poly(dA:dT) transfection which activates the AIM2 inflammasome (Supplementary Fig. 4i).

**NLRP3 S803D does not recruit NEK7.** We next investigated the mechanism of inflammasome blockade by NLRP3 S803 phospho-mimetic substitutions. NEK7 has been shown to be a key component of the NLRP3 inflammasome that interacts and bridges

**Fig. 2 S806 is critical for NLRP3 inflammasome activity in human U937 monocytes.** U937 and NLRP3-deficient U937 cells reconstituted with doxycycline-inducible NLRP3 mutants were sequentially treated with PMA, doxycycline, LPS, and nigericin as indicated. **a** IL-1β and TNF secretions were measured by ELISA. NLRP3 R260W was used as a control of constitutively active mutant. **b** IL-1β, IL-18, and TNF secretions were measured by ELISA. **c** Expression, cleavage, and secretion of IL-1β and caspase-1 were assessed in cell supernatants and lysates by WB. **d** ASC specks were visualized and counted by immunofluorescence (arrows, scale bar 50 μm). Cells transduced with empty lentivector were used as control (−). Quantification of >20 cells per replicates is shown (total cells $n = 408$ pInd, 292 pInd-NLRP3 WT, 377 pInd-NLRP3 S806D). **e** NLRP3-deficient U937 cells reconstituted with doxycycline-inducible NLRP3 WT or S806A mutant were treated with PMA and doxycycline for 16 h, and then with LPS (5 h) and nigericin (20 min). Anti-NLRP3 immunoprecipitates were analyzed for Serine phosphorylation by WB. Means and 1 SD of biological duplicates (**a**, **b**) and 10–11 technical replicates (**d**) are represented. Data are representative of 2 (**c**), 3 (**a**, **e**), and 4 (**b**, **d**) independent experiments. Ordinary two-way ANOVA with Tukey's multiple comparisons of each condition with corresponding WT control, ****$p < 0.0001$. Molecular weights are indicated in kDa (**c**, **d**). pInd pInducer21, Sup supernatants, Lys lysates, IP:NLRP3 anti-NLRP3 immunoprecipitates.

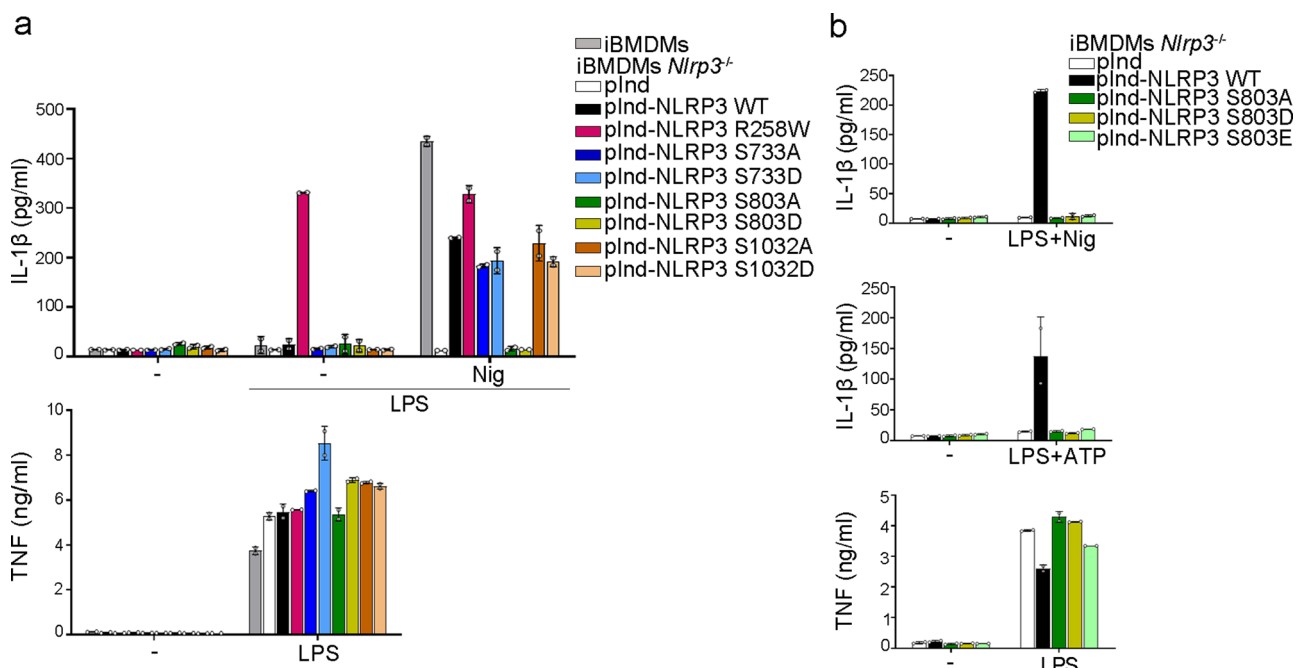

**Fig. 3 S803 is critical for NLRP3 inflammasome activity in immortalized BMDMs. a**, **b** Immortalized WT and $Nlrp3^{-/-}$ BMDMs reconstituted with doxycycline-inducible NLRP3 mutants were sequentially treated with doxycycline, LPS, and nigericin or ATP as indicated. IL-1β and TNF secretions were measured by ELISA. NLRP3 R258W was used as a constitutively active control. Means and 1 SD are represented. Data are biological duplicates representative of 4 (**a**) and 3 (**b**) independent experiment. pInd pInducer21, iBMDMs immortalized BMDMs.

adjacent NLRP3 subunits[13–16]. We confirmed that NEK7 was indeed required for NLRP3 activity in our experimental settings (Supplementary Fig. 5a). As S803 is localized at the interaction surface with NEK7 (Supplementary Fig. 5b), we tested whether S803D substitution may interfere with this recruitment. Indeed, NLRP3 co-immunoprecipitation with NEK7 was abolished in $Nlrp3^{S803D/S803D}$ BMDMs activated with LPS and nigericin (Fig. 5a). We confirmed these results in human U937 monocytes in which phospho-mimetic S806D substitution, but not phospho-null S806A, impaired NEK7 recruitment upon LPS and nigericin treatment (Fig. 5b). NLRP3 S806D substitution similarly blocked NEK7/NLRP3 co-immunoprecipitation in HeLa cells ectopically expressing NLRP3 (Supplementary Fig. 5c). As these over-expression settings bypassed upstream regulations of the inflammasome assembly, we concluded that the phospho-mimetic substitution S806D impacted directly the ability of NLRP3 and NEK7 to interact by modifying the interaction surface. As a control for the correct folding of the LRR domain, we tested NLRP3 S803A, S803D and S803E interaction with the chaperone protein SGT1 known to bind the NLRP3 LRR domain[17]. All three NLRP3 mutants co-immunoprecipitated with VSV-SGT1 when ectopically expressed in 293 T cells excluding any potential misfolding of the LRR domain (Supplementary

Fig. 5d). We next used structure models to investigate the modification of NLRP3/NEK7 interaction surface by S806 phosphorylation (S806p) and S806D phospho-mimetic mutation. Both S806p and S806D mutation, but not S806A mutation, led to modifications of the electrostatic potential surface in the vicinity of S806 (Supplementary Fig. 6a). Noteworthy, electrostatic changes around S806 was also observed in complex models with NEK7 R131E or Q129R mutants known to disrupt NLRP3/NEK7 interaction, suggesting that electrostatic charges may play a role in the stability over time of the different complexes[16]. We further carried out molecular dynamics (MD) trajectories of 100 ns and extracted the number of hydrogen bonds at the end of each simulation. The number of hydrogen bonds in the NLRP3/NEK7 complex was reduced in the case of NLRP3 S806D/NEK7 (9 bonds) similarly as NLRP3/NEK7 Q129R (9 bonds) and NLRP3/NEK7 R131E (11 bonds) used as positive controls of destabilized complex, as compared to NLRP3/NEK7 (15 bonds) or NLRP3 S806A/NEK7 (20 bonds) (Supplementary Table 1). In addition, we measured three distances between residues of NLRP3 and NEK7 respectively connected by hydrogen bonds: $D747_{NLRP3}/R131_{NEK7}$, $R779_{NLRP3}/H125_{NEK7}$ and $E864_{NLRP3}/H125_{NEK7}$, as well as distances of $S806_{NLRP3}$ to its 3 most proximal residues in NEK7 $R121_{NEK7}$, $H125_{NEK7}$ and $K130_{NEK7}$ (Supplementary

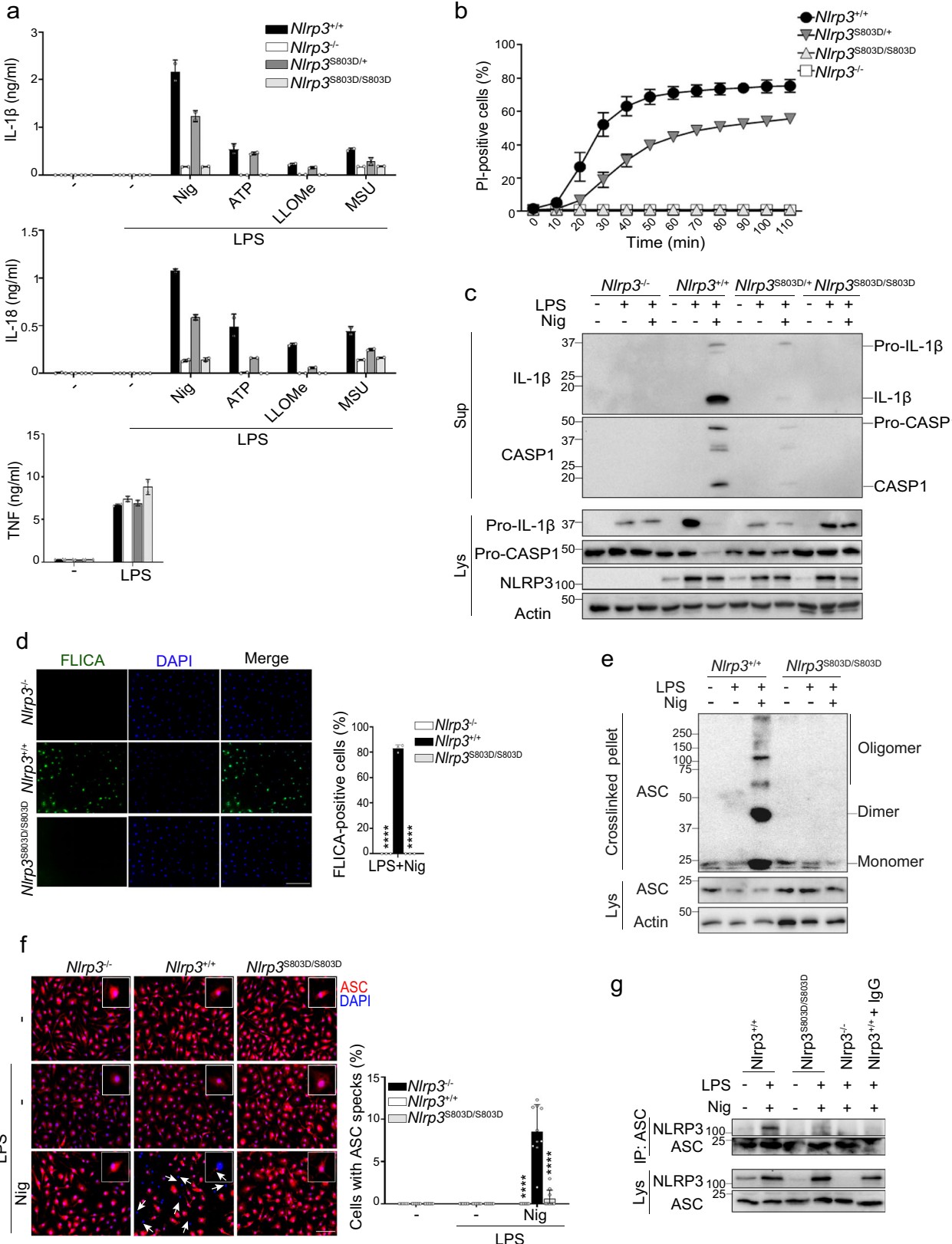

Fig. 6b). As expected, all measured distances in WT NLRP3/NEK7 complex were stable or showed only few variations over time reflecting the stability of the structure, while on the opposite we observed drastic changes over the time with NEK7 Q129R mutant. Similarly, NLRP3 S806D showed important variations of the $E864_{NLRP3}/H125_{NEK7}$, $D747_{NLRP3}/R131_{NEK7}$ and $R779_{NLRP3}/H125_{NEK7}$ distances over time, while these distances were more stable in the NLRP3 S806A/NEK7 complex. To conclude, our structural models suggested that NLRP3 S806D substitution destabilized NEK7 interaction with NLRP3 LRR domain.

**Fig. 4 S803 is critical for NLRP3 inflammasome activity in primary BMDMs. a** BMDMs from *Nlrp3*[S803D/S803D], *Nlrp3*[S803D/+] and *Nlrp3*[+/+] littermate mice were primed with LPS followed by nigericin, ATP, LLOMe, or MSU treatments as indicated. IL-1β, IL-18, and TNF secretions in the supernatant were measured by ELISA. BMDMs from *Nlrp3*[−/−] mice were used as controls. **b** Cell death of LPS-primed BMDMs was monitored by PI incorporation over time following nigericin treatment and quantified by high content microscopy. **c** BMDMs were primed with LPS (6 h) followed by nigericin (30 min) treatment as indicated. Secretion of mature IL-1β and cleaved caspase-1 in the supernatants, and expression of IL-1β, caspase-1, and NLRP3 were assessed by WB. **d** BMDMs were primed with LPS followed by nigericin treatment. Caspase-1 activity was visualized by FLICA staining and confocal microscopy. Scale bar, 50 μm. Quantification of >600 cells per biological triplicates is shown (total cells n = 1966 *Nlrp3*[−/−], 1833 *Nlrp3*[+/+], 2303 *Nlrp3*[S803D/S803D]). **e** BMDMs were treated with LPS and nigericin (30 min). Crosslinked ASC oligomers in the insoluble fraction of cell lysates were visualized by anti-ASC WB. **f** ASC subcellular localization was visualized by fluorescent microscopy (ASC, red; DAPI, blue). Arrow, ASC specks. Scale bar, 20 μm. Quantification of >60 cells per replicates is shown (total cells n = 1005 *Nlrp3*[−/−], 656 *Nlrp3*[+/+], 960 *Nlrp3*[S803D/S803D]). **g** BMDMs were treated with LPS (6 h) and nigericin (30 min). Anti-ASC immunoprecipitates from cell lysates were analyzed for NLRP3 by WB. Lysate of *Nlrp3*[+/+] BMDMs incubated with A/G-beads and isotype control was used as a negative control (*Nlrp3*[+/+]+IgG). Means and 1 SD are represented. Data are biological duplicates representative of three independent experiments (**a**), biological quadruplicates representative of two independent experiment (**b**), biological triplicates representative of two independent experiment (**d**), 10 technical replicates representative of two independent experiments (**f**), one representative of three (**c**, **g**) and four (**e**) independent experiments. Ordinary two-way ANOVA with Tukey's multiple comparisons tests of each condition with corresponding WT control; ****p < 0.0001 (**d**, **f**). Molecular weights are indicated in kDa (**c**, **e**, **g**). Sup supernatants, Lys lysates, IP:ASC anti-ASC immunoprecipitates.

**Impaired NEK7 recruitment blocks BRCC3-mediated NLRP3 deubiquitination.** Since NLRP3 LRR domain recruits the BRCC3 DUB and BRCC3-mediated NLRP3 deubiquitination is critical for the inflammasome assembly, we assessed the impact of S803D phospho-mimetic substitution in BRCC3 recruitment[4]. In *Nlrp3*[S803D/S803D] BMDMs, NLRP3 S803D did not co-immunoprecipitate with BRCC3 upon LPS priming and nigericin activation (Fig. 5c). As already described, NLRP3 was deubiquitinated in LPS-primed BMDMs upon activation with nigericin in a BRCC3-dependent manner[4,18]. On the opposite, NLRP3 S806D mutant got hyper-ubiquitinated in LPS-primed *Nlrp3*[S803D/S803D] BMDMs upon nigericin activation consistently with its impaired BRCC3 recruitment (Fig. 5d). The amount of ubiquitinated NLRP3 S803D increased upon the inhibition of the proteasome by MG132 or lysosomal proteases by E-64d, supporting that NLRP3 S803D hyper-ubiquitination ultimately led to its combined proteasome and lysosome-mediated degradation (Supplementary Fig. 7a). Results were confirmed in human U937 reconstituted with NLRP3 WT, S806A and S806D mutants (Fig. 5e). NLRP3 WT and S806A mutant co-immunoprecipitated with BRCC3 upon LPS and nigericin treatment, while NLRP3 S806D did not. Consistently, NLRP3 WT and S806A mutant were deubiquitinated upon LPS and nigericin treatment, while the NLRP3 S806D mutant was hyper-ubiquitinated (Fig. 5f). Noteworthy, the amount of ubiquitinated NLRP3 S806D further increased upon treatment with MG132 and E-64d confirming that ubiquitinated NLRP3 S806D ultimately got degraded by combined proteasome and lysosome pathways in U937. Noteworthy, NLRP3 protein level was comparable in *Brcc3*[−/−] and control BMDMs (Supplementary Fig. 7b). This observation supported that NLRP3 ubiquitination controlled by BRCC3 was inhibiting NLRP3 activity but did not target WT NLRP3 to degradation, consistently with previous conclusion[4]. Analysis of the ubiquitin chains decorating WT and NLRP3 S803D mutant showed that WT NLRP3 mostly associated with K63 ubiquitin chains in BMDMs treated with LPS and nigericin in the presence of BRCC3 inhibitor G5, while NLRP3 S803D mutant associated with K48 chains (supplementary Fig. 7c). Altogether, these data indicated that WT NLRP3 and NLRP3 S803D mutant differential interactions with partners may lead to different type of ubiquitination and outcome. We next investigated whether BRCC3 and NEK7 recruitment to NLRP3 were linked to one another. BRCC3 was not necessary for NEK7 recruitment as NLRP3 co-immunoprecipitated with NEK7 in *Brcc3*[−/−] BMDMs treated with LPS and nigericin (Supplementary Fig. 7d). Reversely, HA-NEK7 increased NLRP3 co-immunoprecipitation with Myc-BRCC3 when ectopically expressed in 293 T cells (Fig. 5g), and

NLRP3 did not co-immunoprecipitate with BRCC3 in *Nek7*[−/−] immortalized BMDMs treated with LPS and nigericin (Fig. 5h), indicating that NEK7 was required for BRCC3 recruitment to NLRP3. Consistently, the covalent NLRP3 inhibitor oridonin which blocks NEK7 recruitment prevented NLRP3 endogenous co-immunoprecipitation with BRCC3 in BMDMs and led to NLRP3 ubiquitination upon LPS and nigericin treatment (Fig. 5i, j)[19]. Therefore, we concluded that NLRP3 S806D phospho-mimetic mutation prevented the interaction with NEK7 which is necessary for subsequent BRCC3 recruitment and NLRP3 deubiquitination.

***Nlrp3*[S803D/S803D] KI mice have an impaired response to endotoxic shock in vivo.** As NLRP3 was shown to sensitize mice to endotoxic shock, we next assessed the impact of NLRP3 S803D substitution in this in vivo model[20]. Following LPS intraperitoneal injection, *Nlrp3*[S803D/S803D] KI mice showed blunted IL-1β and IL-18 responses in sera and strongly reduced circulating IL-1α which all depend on the inflammasome, while TNF and IL-6 levels in the sera were similar as compared to WT mice (Fig. 6a). Consistent with S803D loss of function, *Nlrp3*[S803D/S803D] KI mice showed increased resistance to LPS intraperitoneal injection similarly to *Nlrp3*[−/−] mice (Fig. 6b and Supplementary Fig. 8).

**CSNK1A1 phosphorylates NLRP3 at S803.** We then investigated the kinase targeting NLRP3 at S806 upon LPS priming. Knockdown (KD) of *Mapk8* encoding JNK1 or its pharmacological inhibition did not reduce IL-1β secretion by LPS and nigericin-treated BMDMs, excluding a role for this kinase in our experimental settings (Supplementary Figs. 9a, b)[12]. In silico analysis using GPS 3.0 and NetPhos 2.0 identified 20 kinases which based on their consensus motifs could phosphorylate NLRP3 at S806 (Supplementary Table 2)[21,22]. SiRNA screen of these candidates revealed that KD of 4 kinase genes (*Csnk1a1*, *Csnk2b*, *Camk4*, *Camk2b*) potently inhibited pyroptosis in BMDMs treated with LPS and nigericin (Fig. 7a and Supplementary Fig. 9c). As *Csnk2b* and *Csnk2a1* encoded respectively the regulatory and catalytic subunits of the Casein Kinase 2 (CSNK2) complex, *Csnk2a1* was also included in the follow-up analysis. SiRNA-mediated KD of *Csnk1a1*, *Camk2b*, *Csnk2b*, and *Csnk2a1* in BMDMs significantly reduced IL-1β secretion following LPS and nigericin treatment (Fig. 7b). In addition, KD of *Csnk1a1* significantly reduced IL-18 secretion and strongly decreased ASC specks formation (Fig. 7c, d). *Csnk1a1* KD did not reduce the expression of *Il1b*, *Nlrp3*, *Pycard* (encoding ASC), *Casp1*, or *Tnf* at the mRNA and protein levels, excluding that CSNK1A1 was required for LPS-triggered

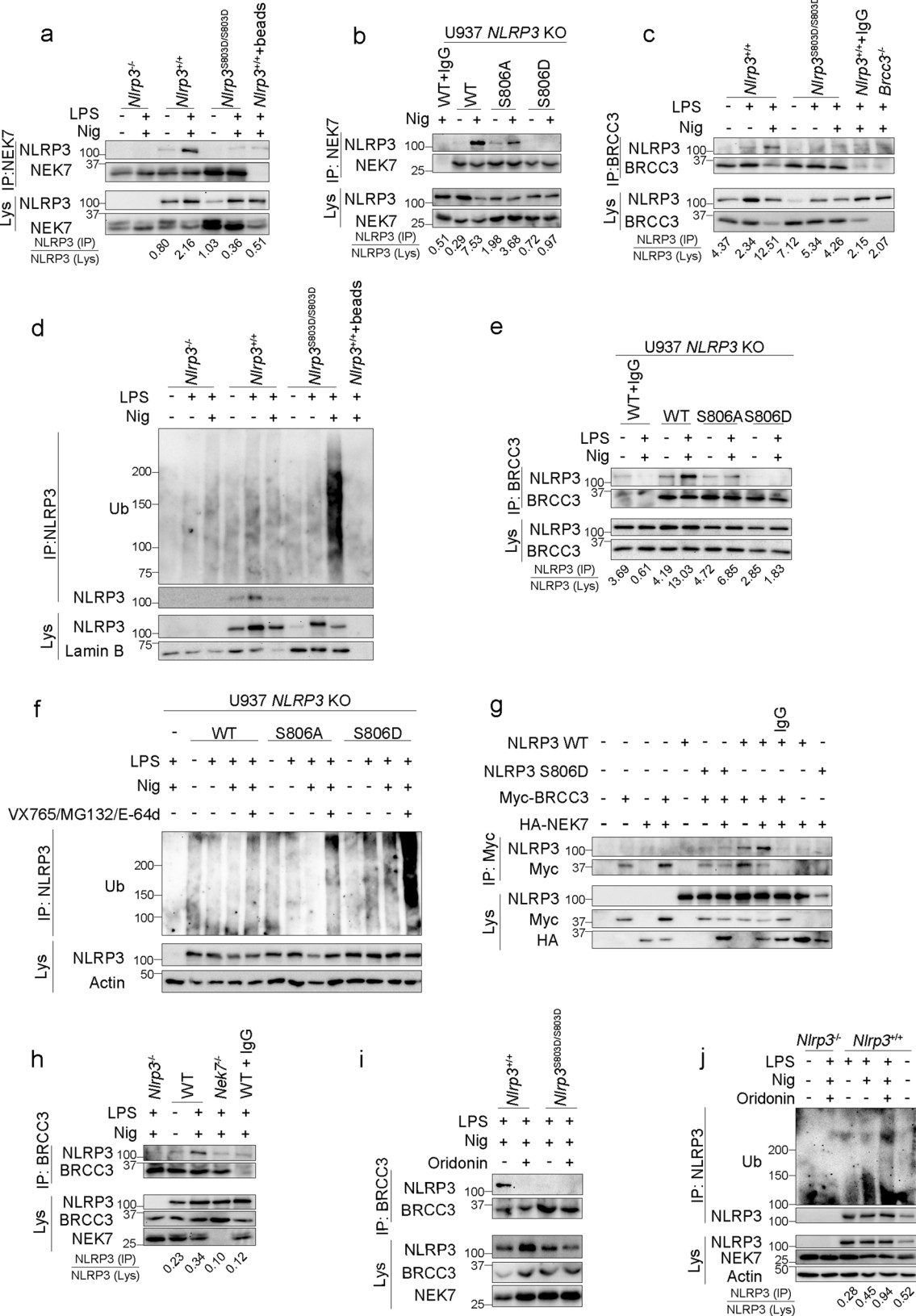

transcriptional response (Supplementary Fig. 9d, e). In in vitro kinase assay, recombinant Casein Kinase 1 alpha1 (CSNK1A1) phosphorylated GST-NLRP3, while GST-NLRP3 incubation with CSNK2A1/CSNK2B or CAMK2B resulted in very low and no phosphorylation of GST-NLRP3 respectively (Fig. 7e). NLRP3 co-immunoprecipitated with HA-CSNK1A1 when ectopically expressed in 293T cells (Fig. 7f), and endogenous NLRP3 co-immunoprecipitated with CSNK1A1 in BMDMs primed with LPS (Fig. 7g). Finally, HA-CSNK1A1 co-expression with NLRP3 in 293T cells resulted in the phosphorylation of NLRP3 WT but not NLRP3 S806A mutant (Fig. 7h). Consistently, CSNK1A1 inhibitor D4476 completely abrogated NLRP3 phosphorylation on serine in

**Fig. 5 S803D phospho-mimetic mutation blocks NEK7 recruitment by NLRP3. a** Endogenous NEK7 immunoprecipitates from LPS-primed BMDMs treated with nigericin (30 min) were analyzed for NLRP3 (*Nlrp3*$^{+/+}$+beads, *Nlrp3*$^{+/+}$ BMDM lysate incubated with A/G-beads without anti-NEK7). **b** NLRP3-deficient U937 cells reconstituted with NLRP3 mutants were treated with PMA, doxycycline, LPS, and nigericin. Endogenous NEK7 immunoprecipitates were analyzed for NLRP3 (WT + IgG, lysate of U937 expressing WT NLRP3 incubated with isotype control and A/G-beads). **c, d** BMDMs were treated with LPS (6 h) and nigericin (30 min). Endogenous BRCC3 immunoprecipitates were analyzed for NLRP3 (*Nlrp3*$^{+/+}$+IgG, BMDM lysate incubated with IgG isotype control and A-beads; *Brcc3*$^{-/-}$, BRCC3 immunoprecipitates from *Brcc3*$^{-/-}$ BMDM lysate) (**c**). NLRP3 ubiquitination was assessed by NLRP3 immunoprecipitation followed by anti-Ub WB (*Nlrp3*$^{+/+}$+beads, *Nlrp3*$^{+/+}$ BMDM lysate incubated with A/G-beads without anti-NLRP3) (**d**). **e** NLRP3-deficient U937 cells reconstituted with NLRP3 mutants were treated with PMA and doxycycline (2 μg/ml, 16 h) followed by LPS (5 h) and nigericin (20 min). Endogenous BRCC3 immunoprecipitates were analyzed for NLRP3 (WT + IgG, lysate of U937 cells reconstituted with NLRP3 WT incubated with isotype control and A-beads). **f** NLRP3-deficient U937 cells reconstituted with NLRP3 mutants were treated with PMA and doxycycline (2 μg/ml, 16 h) followed by LPS (50 ng/ml, 4 h) and MG132 (10 μM, 40 min), E-64d (20 μg/ml, 40 min) and VX765 (2.5 μM, 40 min) 10 min before nigericin (30 min). NLRP3 ubiquitination was assessed by NLRP3 immunoprecipitation followed by anti-Ub WB. Caspase-1 inhibitor VX765 was added to prevent pyroptosis. **g** NLRP3 WT and S806D mutant were expressed with Myc-BRCC3 in 293T cells in the presence or not of HA-NEK7. Myc-BRCC3 immunoprecipitates were analyzed for NLRP3 (IgG, lysate of 293T cells expressing NLRP3 WT, Myc-BRCC3 and HA-NEK7 incubated with isotype control and A/G-beads). **h** Immortalized WT and *Nek7*$^{-/-}$ BMDMs were treated with LPS (6 h) and nigericin (30 min) in the presence of VX765 (2.5 μM, 15 min before nigericin). BRCC3 immunoprecipitates were analyzed for NLRP3 by WB (WT + IgG, lysates of immortalized WT BMDMs incubated with isotype control and A-beads). **i, j** BMDMs were treated with LPS (6 h) and nigericin (30 min) with oridonin (2 μM, 30 min before nigericin). Endogenous BRCC3 immunoprecipitates were analyzed for NLRP3 (**i**). NLRP3 ubiquitination was assessed by NLRP3 immunoprecipitation followed by anti-Ub WB (**j**). Data are one representative of three independent experiments. Molecular weights are indicated in kDa. Lys lysates, IP immunoprecipitates, Ub ubiquitin.

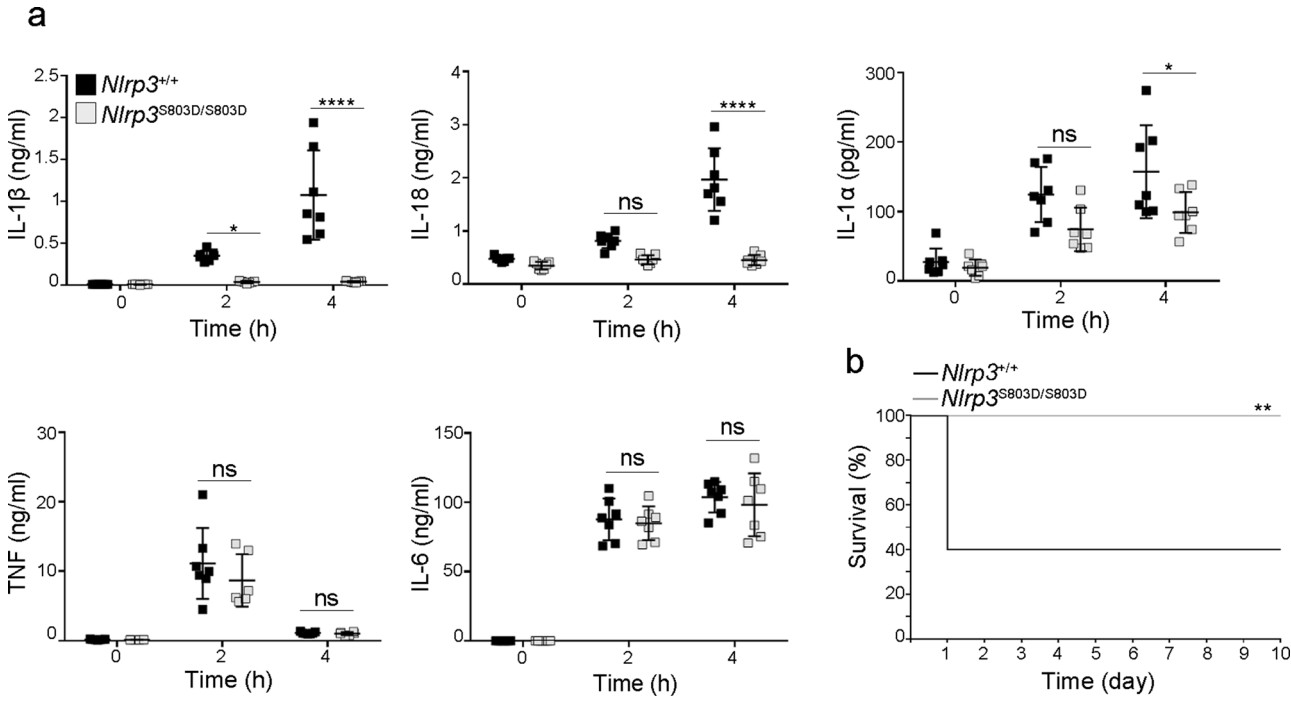

**Fig. 6 Nlrp3$^{S803D/S803D}$ KI mice show an impaired response to endotoxic shock in vivo.** *Nlrp3*$^{S803D/S803D}$ and *Nlrp3*$^{+/+}$ littermates were injected intraperitoneally with LPS. **a** Indicated cytokines were measured in the sera collected before as well as 2 and 4 h post-injection. *Nlrp3*$^{S803D/S803D}$ and *Nlrp3*$^{+/+}$ (*n* = 7). Individual values, means and 1 SD are represented, repeated measure two-way ANOVA with Sidak's multiple comparisons of each condition with corresponding WT control; ns, non significant; *$p$-value < 0.05; ****$p$-value < 0.0001. **b** Mice survival to endotoxic shock. *Nlrp3*$^{S803D/S803D}$ (*n* = 11), *Nlrp3*$^{+/+}$ (*n* = 10), two-sided Mantel-Cox test, **$p$-value < 0.01. Data are biological replicates of one experiment.

LPS-primed U937 (Fig. 7i). Altogether these results identify CSNK1A1 as the kinase responsible for NLRP3 phosphorylation at S803 following LPS priming.

## Discussion

NLRP3 post-translational modifications recently emerged as a major control process of the inflammasome assembly. In particular, NLRP3 LRR deubiquitination by BRCC3 plays a key role for inflammasome assembly[4,18]. Using a biochemical approach, we identified here S803 in the LRR domain of mouse NLRP3 (corresponding to S806 in human NLRP3) to be critical for BRCC3 recruitment. NLRP3 is phosphorylated at S803 upon priming and dephosphorylated upon activation. S803 is critical for inflammasome activity in both human monocytes and murine macrophages regardless of the nature of the priming and activation signals, suggesting that S803 phosphorylation status is a universal checkpoint for inflammasome assembly. Consistently, *Nlrp3*$^{S803D/S803D}$ KI mice are resistant to endotoxic shock in vivo. Mechanistically, S803D substitution impairs the recruitment of NEK7 to NLRP3 by modifying their interaction surface. We evidenced this recruitment to be a prerequisite for BRCC3 binding. Subsequently, NLRP3 S803D is hyper-ubiquitinated resulting in its proteasome and lysosome-mediated degradation. Finally, we identified CSNK1A1 as the critical kinase targeting NLRP3 at S803. Altogether, our results highlight S803

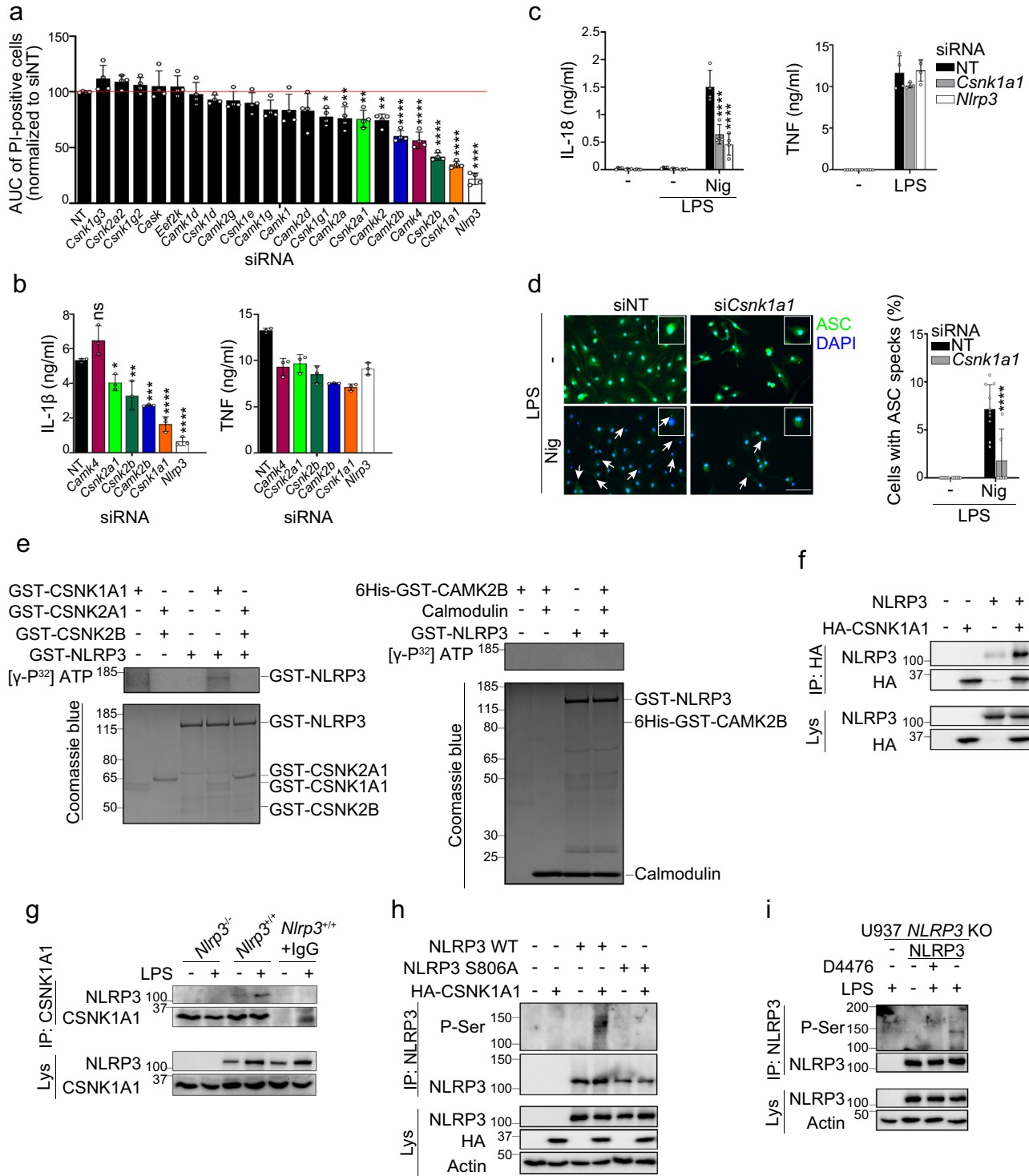

phosphorylation status as a critical checkpoint of inflammasome activation.

S803 is phosphorylated upon priming and dephosphorylated upon activation, and both phospho-null S803A and phospho-mimetic S803D/S803E substitutions impair NLRP3 inflammasome assembly. These phenotypes are reminiscent of substitutions of phosphorylated S291 (in mouse NLRP3 corresponding to S295 in human NLRP3), with both S291A and S291E being inactive[23]. Mechanistic insights revealed that NLRP3 S291 dephosphorylation is required for NLRP3 initial recruitment in the mitochondria-associated membranes (MAMs) where

inflammasome probably assembles initially, while NLRP3 phosphorylation by Golgi-associated PKD is later required for inflammasome release from the MAMs to the speck. Thus, the consequences of S291 phosphorylation vary in a timely and localized manner, reconciling apparently conflicting studies[23–25]. Similarly, both phospho-null and phospho-mimetic mutants of S3 (in mouse NLRP3, corresponding to S5 in human NLRP3) are inhibitory. Indeed S3 phosphorylation by AKT transiently protects NLRP3 from degradation upon inflammasome priming, while it blocks interaction with ASC upon speck formation[26,27]. Similarly, NLRP3 phosphorylation of S803 must require a timely regulation

**Fig. 7 CSNK1A1 phosphorylates NLRP3 on S806. a** BMDMs transfected with the indicated siRNAs were treated with LPS followed by nigericin. Cell death was monitored by PI incorporation over time (for 140 min following nigericin treatment) and quantified by high content microscopy. Means of area under the curve normalized to non-targeting siRNA control and 1 SD are represented. **b** IL-1β and TNF secretions were measured by ELISA. **c, d** BMDMs transfected with *Csnk1a1* siRNA were treated with LPS (6 h) followed by nigericin. IL-18 and TNF secretions were measured by ELISA (**c**). ASC specks were visualized and counted by immunofluorescence (**d**). Arrow, ASC specks. Scale bar, 20 μm. Quantification of >10 cells per replicates is shown (total cells $n = 571$ NT, 158 *Csnk1a1*). Means and 1 SD are represented. **e** GST-NLRP3 was incubated with GST-CSNK1A1, GST-CSNK2A1/GST-CSNK2B, or 6-His-GST-CAMK2B. In vitro phosphorylation was revealed by SDS-PAGE and autoradiography. Coomassie stainings serve as controls. **f** NLRP3 was ectopically expressed with HA-CSNK1A1 in 293T. HA-CSNK1A1 immunoprecipitates were analyzed for NLRP3. **g** BMDMs were treated with LPS (6 h). CSNK1A1 immunoprecipitates were analyzed for NLRP3. Lysate of $Nlrp3^{+/+}$ BMDMs incubated with isotype control and A/G-beads ($Nlrp3^{+/+}$+IgG) and lysates of $Nlrp3^{-/-}$ BMDMs were used as negative controls. **h** NLRP3 WT or S806A mutant were ectopically expressed with HA-CSNK1A1 in 293 T. NLRP3 immunoprecipitates were analyzed for phospho-Ser by WB. **i** NLRP3-deficient U937 cells reconstituted with NLRP3 were treated with PMA and doxycycline for 16 h followed by D4476 and LPS (4 h). NLRP3 immunoprecipitates were analyzed for phospho-Ser by WB. Data are biological quadruplicates representative of two independent experiment (**a, c**), biological triplicates representative of four independent experiment (**b**), 11 technical replicates representative of two independent experiments (**d**), one representative of two (**e, h, i**) and three (**f, g**) independent experiments. Ordinary one-way ANOVA with Dunnett's multiple comparisons (**a**), ordinary two-way ANOVA with Tukey's multiple comparisons (**c, d**) to corresponding non-targeting siRNA control; *p-value < 0.05; **p-value < 0.01; ***p-value <0.001; ****p-value < 0.0001. Molecular weights are indicated in kDa (**e–i**). AUC area under the curve, NT non-targeting siRNA, Lys lysates, IP immunoprecipitates.

and both phosphorylation upon priming and dephosphorylation upon activation are sequentially required for the inflammasome assembly. Defective inflammasome functions in cells expressing NLRP3 S803A mutants is consistent with the inhibitory phenotype caused by the KD of CSNK1A1 kinase expression, which we identify to phosphorylate NLRP3 at S803. As NLRP3 S803A mutation completely abolishes NLRP3 Serine phosphorylation following LPS priming, NLRP3 phosphorylation at S803 may be a prerequisite for additional priming-associated phosphorylation required for rendering NLRP3 competent for further activation[12]. Interestingly, NLRP3 S803A mutant recruits NEK7 and BRCC3 upon activation, although it does not restore inflammasome activity. These observations support that NEK7 recruitment is necessary but not sufficient for active inflammasome assembly, which depends on additional S803 phosphorylation-dependent regulatory mechanisms. Noteworthy, human NLRP3 phospho-null S806A mutant shows residual activity in human U937 cells, suggesting that priming-associated NLRP3 phosphorylation at S806 may be partially dispensable in this cell line, and confirming species- and cell type-dependent regulation of the inflammasome as already described[28,29].

NLRP3 is dephosphorylated at S803 upon activation signal. NLRP3 S803D complete loss of function phenotype indicates that this dephosphorylation is mandatory for inflammasome assembly. We evidence that NLRP3 S803A but not S803D mutants co-immunoprecipitates with NEK7. Indeed S803 is located at the interaction site with NEK7 and according to our in silico models its phosphorylation affects electrostatic potential and hydrogen bonds involved in this interaction surface and reduces the stability of the NLRP3/NEK7 complex[16]. Impairment of NLRP3 pS803-NEK7 binding provides a molecular mechanism for the absolute requirement of S803 dephosphorylation. Although LRR domain has been reported to be dispensable for NLRP3 inflammasome regulation and assembly, no evidence in endogenous context have been provided[30]. In addition, we evidence that LRR truncation of NLRP3 is inactive in two reconstitution systems matching endogenous expression level, independently of slight variations in the protocols used for priming and activation with this study[30]. The NLRP3 pS803 regulatory mechanism is reminiscent of NLRP3 phosphorylation at Y859 also located in this interaction surface and the phosphorylation of which inhibits NLRP3-NEK7 interaction by steric hindrance and charge repulsion[16]. Therefore, in the widely used experimental model of LPS transcriptional priming for >3 h followed by activation, activation-triggered NLRP3 S803 dephosphorylation is a prerequisite for NLRP3 binding to NEK7 which occurs following the

activation signal in this system[13,14]. Recent work suggests that NEK7 may be dispensable for inflammasome activity in human and murine cells in some specific conditions of priming and activation[31]. Given the proposed mechanism, we assume that NLRP3 dephosphorylation at S803 might only affect NEK7-dependent inflammasome activation. However, we cannot exclude that S803 site may be involved in interaction with other partner(s) or regulator(s) and may be critical for NEK7-independent inflammasome activation as well.

In addition to NEK7, NLRP3 S803D phospho-mimetic mutant is also impaired in the recruitment of the BRCC3 DUB upon activation. While overexpression of NEK7 increases the recruitment of BRCC3 to NLRP3 when ectopically expressed, NEK7 deficiency or oridonin-based chemical disruption of NLRP3-NEK7 binding block it. Therefore, NLRP3-NEK7 binding appears to be a prerequisite for BRCC3 recruitment. Noteworthy, as no tripartite complex could be observed, NLRP3-NEK7 and NLRP3-BRCC3 interactions might be sequential. BRCC3 binds to ubiquitinated NLRP3 LRR[4], but the critical E3 ligase still remains to be identified. Three E3 are known to ubiquitinate the LRR domain: Pellino2 participates in inflammasome priming, while MARCH7 and RNF125 are inhibitory[5,7,8]. We show here that the defect in BRCC3 recruitment by NLRP3 S803D mutant ultimately leads to its hyper-ubiquitination and its degradation. Noteworthy, RNF125-mediated ubiquitination of the LRR domain with K63 chains controls the sequential recruitment of E3 CBL-B that ubiquitinates NLRP3 NACHT domain with K48 chains leading to its downregulation by proteasome-mediated degradation[5]. Whether RNF125-mediated LRR ubiquitination may control BRCC3 recruitment, or whether RNF125 and BRCC3 may antagonize each other for CBL-B recruitment would require further investigations.

We identify CSNK1A1 as the kinase targeting NLRP3 at S803. Indeed, KD of CSNK1A1 was the most potent hit of our siRNA-based screen for inflammasome modulators among a selection of kinase candidates with consensus sites matching NLRP3 S803 surrounding sequence. Phosphorylation of NLRP3 at S803 by CSNK1A1 was confirmed by in vitro as well as in cellulo kinase assays. CSNK1A1, also referred to as CK1α, is part of the CSNK1 monomeric Ser/Thr kinase family comprising 7 members (CSNK1A1, CSNK1B, CSNK1D, CSNK1E, CSNK1G1, CSNK1G2, CSNK1G3) encoded by distinct genes. CSNK1s present highly conserved kinase domains and diverse C-terminal domains. CSNK1A1 is ubiquitously expressed in various organs. Its consensus target site corresponds to pS/pT/D/E-X$_{1-2}$-S/T matching NLRP3 $\underline{D}_{801}L_{802}\underline{S}_{803}$ sequence. CSNK1A1 has

numerous identified substrates including β-catenin, p53, PMRT1, IFNAR1, AGO2, FOXO3A, p62, UVRAG and RIPK3. CSNK1A1 regulates various biological functions and conditions including development, cancer, circadian cycle, membrane transportation, immune response, neurodegeneration, autophagy, cell cycle, RNA processing, phagocytosis and cell death[32–37]. Consistently, *Csnk1a1* knock-out is embryonically lethal. However, our study reports for the first time a function of CSNK1A1 in the regulation of inflammasome. Although CSNK1A1 is described to be constitutively active, inhibitory autophosphorylation of its C-term domain suggests regulation of this activity, which would require further investigation in the inflammasome activation context. Future development of our work would consider CSNK1A1 potential as a druggable target to inhibit NLRP3 inflammasome activation against highly prevalent inflammatory conditions and inflammaging. Although CSNK1A1 pleiotropic roles suggest possible side effects of its inhibition that may limit the use of inhibitors of its activity to acute inflammatory flares and exclude long term treatment of chronic inflammatory conditions, development of inhibitors targeting CSNK1A1 binding to NLRP3 should provide highly specific inhibitor of the NLRP3 pathway.

## Methods

**Cell culture**. 293 T cells, immortalized BMDMs and HeLa cells were cultured at $0.5–1 \times 10^6$ cells/ml in Dulbecco's modified Eagle's Medium (DMEM) Gluta-Max$^{TM}$-I supplemented with 1X penicillin/streptomycin (PS) and 10% Fetal Bovine Serum (FBS) (Gibco). Immortalized BMDMs from C57Bl6J WT and *Nlrp3*$^{-/-}$ mice were a kind gift from Pr E. Alnemri (Jefferson University, Philadelphia). Immortalized BMDMs from C57Bl6J WT and *Nek7*$^{-/-}$ mice were a kind gift from Pr G. Nuñez (University of Michigan, Ann Arbor, USA). U937 cells were cultured at $0.5–1 \times 10^6$ cells/ml in Roswell Park Memorial Institute (RPMI) 1640 GlutaMax$^{TM}$-I supplemented with 1X penicillin/streptomycin (PS) and 10% Fetal Bovine Serum (FBS) (Gibco) and differentiated in macrophages with Phorbol-12-myristate-13 acetate (PMA) (Invivogen) (50 ng/ml, 16 h) before each experiment. NLRP3-deficient U937 have been previously described[10]. Primary BMDMs were differentiated from bone marrow progenitors in DMEM GlutaMax$^{TM}$-I supplemented with 10% FBS (Gibco), 10% M-CSF producing NIH 3T3 conditioned medium, 1% Hepes (Life technologies), 1% Na-pyruvate (Life technologies) for 6 days. For analysis of caspase-1 and IL-1β secretion by WB, cell media were replaced by Opti-MEM medium (Gibco) prior treatments.

**Plasmids**. Constructs coding for Flag-NLRP3 and Flag-NLRP3-LRR were a kind gift from F. Martinon (Unil, Lausanne, Switzerland), VSV-SGT1 from V. Petrilli (CRCL, Lyon, France), Myc-BRCC3 from W. Harper (Harvard Medical School, Boston), pcDNA3.1-HA-CSNK1A1 from P. Wang (Tongji University, Shanghai); pcDNA3-N-HA-NEK7 was purchased from Addgene. Human NLRP3 (hNLRP3), mouse NLRP3 (mNLRP3), and truncation deletions were cloned in pENTR$^{TM}$1A (Invitrogen). Mutations were performed using QuickChange$^{TM}$ II kit (Agilent Technologies). Primers are listed in Supplemental Table 3. cDNAs were transferred in pInducer21 using recombination Gateway LR clonase Enzyme mix kit (Thermofisher)[38].

**Reagents**. The following reagents were used: MG132 (Sigma-Aldrich), E-64d (Enzo Life Science), VX765 (Invivogen), G5 (Merck), doxycycline (Sigma-Aldrich), LPS (O111:B4, Sigma-Aldrich), Pam2CSK4 (Invivogen), nigericin (Invivogen), ATP (Sigma-Aldrich), H-Leu-Leu-OMe (LLOMe, Santa Cruz Biotechnology), MSU (Invivogen), silica (U.S.Silica), poly(dA:dT) (Invivogen), flagellin (FLA-ST Invivogen), oridonin (Euromedex), and D4476 (Sigma-Aldrich), SP600125 (Selleck Chemical).

**Cell transfection**. 293T cells were transfected by calcium chloride methods. U937, immortalized BMDMs and HeLa cells were transfected with poly(dA:dT), flagellin or plasmids using Lipofectamine 2000 (Invitrogen) according to the manufacturer's instructions. For siRNA, BMDMs were transfected with SMARTpools of 4 siRNAs (Dharmacon) using Lipofectamine RNAiMax (Invitrogen) for 2 days.

**Cell transduction**. NLRP3-deficient U937 cells and *Nlrp3*$^{-/-}$ immortalized BMDMs were transduced with lentiviruses generated using pInducer21-hNLRP3 or pInducer21-mNLRP3 and their indicated mutants, respectively. Transduced GFP-positive cells were sorted by cytometry as bulk (Supplementary Fig. 10).

**Immunoprecipitation**. For Flag-LRR purification, 293T cell lysates in RIPA buffer (50 mM Tris-HCl pH 8.0, 300 mM NaCl, 1 mM EDTA, 0.05% Sodium deoxycholate, 1% Triton X-100, 0.1% SDS, 1 mM PMSF, 1 mM NaF, 1 mM Na$_3$VO$_4$, and EDTA-free protease inhibitor cocktail (Roche)) were incubated with anti-Flag M2 affinity agarose gel (Sigma-Aldrich) for 16 h. Beads were washed 15 times in RIPA buffer, and Flag-LRR was eluted in RIPA buffer with 100 µg/ml 3XFlag peptide (Sigma-Aldrich). For NLRP3 ubiquitination assay, lysates of $2 \times 10^6$ BMDMs (10 mM Tris-HCl pH 8, 300 mM NaCl, 2% SDS, 50 mM NaF, 2 mM Na$_3$VO$_4$, EDTA-free protease inhibitor cocktail), were boiled for 10 min then diluted 10X in dilution buffer (10 mM Tris-HCl pH 8, 150 mM NaCl, 1 mM EDTA, 1% Triton X-100, EDTA-free protease inhibitor cocktail). Cleared lysates were incubated with anti-NLRP3 and Protein A/G Ultralink$^{TM}$ resin (Thermo Scientific) for 16 h. Beads were washed 10 times with wash buffer (10 mM Tris-HCl pH 8, 300 mM NaCl, 0.1% SDS, 0.05% Sodium deoxycholate, 1% Triton X-100, 1 mM PMSF, 50 mM NaF, 2 mM Na$_3$VO$_4$, EDTA-free protease inhibitor cocktail), and eluted in sample buffer 2X (250 mM Tris-HCl pH 6.8, 25% Glycerol, 5% SDS, 10% β-Mercaptoethanol, Bromophenol Blue). For endogenous BRCC3 co-IP, lysates of $1.2 \times 10^6$ BMDMs or $4 \times 10^6$ U937 cells (50 mM Tris-HCl pH 7.4, 150 mM NaCl, 2 mM EDTA, 1% NP-40, 1 mM PMSF, 1 mM NaF, 1 mM Na$_3$VO$_4$, EDTA-free protease inhibitor cocktail) were incubated with anti-BRCC3 (Cell Signaling Technology) or IgG (Diagenode) for 16 h, and ProteinA-Dynabeads® (Thermofisher) for 3 h. For ASC co-IP, $6 \times 10^6$ BMDMs were treated with VX765 (2.5 µM) 15 min before activation by nigericin and then mechanically lysed in hypotonic buffer (20 mM Hepes pH 7.4, 10 mM KCl, 1 mM PMSF, 1 mM NaF, 1 mM Na$_3$VO$_4$, EDTA-free protease inhibitor cocktail) by 50 strokes through 25G syringe. Lysates were cleared by centrifugation (350 g 5 min 4 °C), diluted in 2X IP buffer (100 mM Tris pH 7.4, 300 mM NaCl, 0.2% NP-40, 20 mM MgCl$_2$, 1 mM PMSF, 1 mM NaF, 1 mM Na$_3$VO$_4$, EDTA-free protease inhibitor cocktail) containing 25 U/ml benzonase (Sigma-Aldrich), and incubated with anti-ASC (AL177, Adipogen) for 16 h and ProteinA-Dynabeads® for 1 h. For ectopically expressed NLRP3 co-IP, $6 \times 10^6$ HeLa cells were transfected with pcDNA3-NLRP3 WT, S803A or S803D mutants. 30 h later, lysates (100 mM Tris-HCl pH 8, 90 mM NaCl, 10 mM MgCl$_2$, 0.1% Triton X-100, EDTA-free protease inhibitor cocktail) were incubated with anti-NLRP3 and Protein G-Sepharose beads (Sigma-Aldrich) for 16 h. For endogenous NEK7 co-IP, lysates of $3 \times 10^6$ BMDMs or $3.6 \times 10^6$ U937 cells (50 mM Tris-HCl pH 7.4, 150 mM NaCl, 2 mM EDTA, 1% NP-40, 1 mM PMSF, 1 mM NaF, 1 mM Na$_3$VO$_4$, EDTA-free protease inhibitor cocktail) were incubated with anti-NEK7 (EPR4900, Abcam) and Protein A/G resin for 16 h. For VSV-SGT1 co-IP, $1.2 \times 10^6$ 293T cells were transfected with plasmid coding for VSV-SGT1, and pcDNA3-NLRP3 WT, S803A, S803D or S803E mutants. 24 h later, cells were lysed in NP-40 lysis buffer (50 mM Tris-HCl pH 7.4, 150 mM NaCl, 2 mM EDTA, 1% NP-40, 1 mM PMSF, 1 mM NaF, 1 mM Na$_3$VO$_4$, EDTA-free protease inhibitor cocktail) and lysates were incubated with anti-VSV (P5D4, Sigma-Aldrich) and Protein A/G resin for 16 h. For Myc-BRCC3 co-IP, $1.2 \times 10^6$ 293T cells were transfected with plasmid coding for Myc-BRCC3, pcDNA3-NLRP3 WT or S803D, and pcDNA3-N-HA-NEK7. 24 h later, lysates (50 mM Tris-HCl pH 7.4, 150 mM NaCl, 2 mM EDTA, 1% NP-40, 1 mM PMSF, 1 mM NaF, 1 mM Na$_3$VO$_4$, EDTA-free protease inhibitor cocktail) were incubated with anti-Myc (C3956, Sigma-Aldrich) and Protein A/G resin for 16 h. For HA-CSNK1A1 co-IP, $1.2 \times 10^6$ 293T cells were transfected with pcDNA3.1-HA-CSNK1A1 and pcDNA3-NLRP3 WT. 24 h later, lysates (50 mM Tris-HCl pH 7.4, 150 mM NaCl, 2 mM EDTA, 1% NP-40, 1 mM PMSF, 1 mM NaF, 1 mM Na$_3$VO$_4$, EDTA-free protease inhibitor cocktail) were incubated with anti-HA-agarose beads (A2095, Sigma-Aldrich) for 16 h. For endogenous CSNK1A1 co-IP, $6 \times 10^6$ BMDMs were treated with LPS (50 ng/ml, 6 h). Lysates (150 mM Tris-HCl pH 7.4, 300 mM NaCl, 2 mM EDTA, 1% NP-40, 1 mM PMSF, 1 mM NaF, 1 mM Na$_3$VO$_4$, EDTA-free protease inhibitor cocktail) were incubated with anti-CSNK1A1 (301–991 A, Bethyl Laboratories) and Protein A/G resin for 16 h. For all co-IPs, beads were washed six times in lysis buffer and eluted in sample buffer 2X. Antibodies are listed in Supplemental Table 4.

**Mass spectrometry**. Flag-LRR mass spectrometry analysis was performed at the Taplin facility (R. Tomaino and S. Gygi, Harvard Medical School, Boston). The samples were reduced (1 mM DTT, 30 min 60 °C), alkylated (5 mM iodoacetamide, 15 min) and in-gel digested (50 mM ammonium bicarbonate solution containing 12.5 ng/µl trypsin (Promega), 16 h 37 °C,). Peptides were extracted, washed (50% acetonitrile, 1% formic acid), dried and reconstituted (2.5% acetonitrile, 0.1% formic acid). Samples were loaded via a Famos auto sampler (LC Packings) onto equilibrated nano-scale reverse-phase HPLC capillary columns containing 2.6 µm C18 spherical silica beads into a fused silica capillary (100 µm inner diameter x ~30 cm length)[39]. Peptides were eluted with increasing concentrations of solvent (97.5% acetonitrile, 0.1% formic acid), subjected to electrospray ionization and entered into an LTQ Orbitrap Velos Pro ion-trap mass spectrometer (Thermo Fisher Scientific). Eluted peptides were fragmented to produce a tandem mass spectrum of specific fragment ions. Sequences were determined by matching databases with the fragmentation pattern using Sequest (ThermoFinnigan). The modifications of 79.9663 mass units to serine, threonine, and tyrosine, and of 114.0429 Da to lysine were included in the database searches to determine phospho- and ubiquitinated peptides respectively. Phosphorylation assignments were determined by the Ascore algorithm[40]. All databases include a reversed version of

all the sequences and the data was filtered to between 1–2% peptide false discovery rate.

**Inflammasome activation.** Inflammasome was activated using the following protocols unless stated otherwise. $0.5 \times 10^6$ U937 cells/ml were differentiated with PMA (50 ng/ml, 16 h), treated with doxycycline (1 μg/ml, 4 h), primed with LPS (40–50 ng/ml, 3–4 h) or Pam3CSK4 (0.5 μg/ml), and activated with nigericin (15 μg/ml, 1 h), MSU (250 μg/ml, 16 h) or silica (1 mg/ml, 16 h), or transfected with poly(dA:dT) (1 μg/ml, 6 h). $0.3 \times 10^6$ immortalized BMDMs cells/ml were treated with doxycycline (2 μg/ml, 4 h), primed with LPS (50 ng/ml, 4 h), and activated with nigericin (15 μg/ml, 2 h), ATP (2 mM, 30 min) or transfected with flagellin (10 μg/ml, 6 h). $0.3 \times 10^6$ primary BMDMs/ml were primed with LPS (50 ng/ml, 4 h) or Pam3CSK4 (1 μg/ml, 4 h), and activated by addition of nigericin (15 ug/ml, 1 h), ATP (2 mM, 30 min), LLOMe (1 mM, 1 h) or MSU (250 μg/ml, 16 h) to the media, or transfected with flagellin (10 μg/ml, 6 h), poly(dA:dT) (1 μg/ml, 6 h) or LPS (500 ng/ml, 20 h) with Lipofectamine 2000 (ThermoFisher) according to the manufacturer's instructions. For bacterial infection, U937 and immortalized BMDMs were grown in the absence of PS, S. *Typhimurium* SL1344 (MOI 2.4) were added to the media for 2 h. Primary BMDMs were grown in the absence of PS and primed with LPS (50 ng/ml, 4 h) before addition of S. *Typhimurium* SL1344 (MOI 234) for 2 h, or addition of E. *coli* J53 (MOI 100) for 16 h with gentamycin (10 μg/ml) added 1 h after E. *coli* J53 infection.

**ELISA.** The following ELISA kits were used: human IL-1β, human TNF, mouse IL-1β, mouse TNF (Duoset, RnD Systems), human IL-18 (MBL), mouse IL-18 (Thermofisher). Data were acquired using i-control (Tecan). Measured values are available in the Source Data file. Cytokines in sera were measured using the following Luminex kits: mouse IL-6, mouse IL-1a, mouse IL-1b, mouse TNF, mouse IL-18 (Bio-Rad), and analyzed with Luminex Bioplex manager (Bio-Rad).

**Western-blot analysis.** To test protein level in lysates, cells were directly lysed in sample buffer 2X. To test protein secretion, cell supernatants were concentrated using Amicon Ultra-0.5 mL Centrifugal Filters (Millipore) and diluted in sample buffer 2X. Samples were analyzed by SDS-PAGE and transferred to PVDF membranes. The following antibodies were used: anti-Flag (M2, Sigma-Aldrich), anti-Ub (Dako, Cell Signaling Technology and Santa Cruz Biotechnology), anti-K48 Ub (4289, Cell Signaling Technology), anti-K63 Ub (5621, Cell Signaling Technology), anti-IL-1β (166926, RnD Systems, and D3U3E, Cell Signaling Technology), anti-Caspase-1 (5B10, BioLegend), anti-Actin (C4, Sigma-Aldrich), anti-NLRP3 (Cryo2, Adipogen), anti-ASC (Santa Cruz Biotechnology and Adipogen), anti-BRCC3 (Cell Signaling Technology), anti-LaminB (B-10, Santa Cruz Biotechnology), anti-HA (HA-7, Sigma-Aldrich), anti-VSV (P5D4, Sigma-Aldrich), anti-NEK7 (EPR4900, Abcam), anti-Myc (C3956, Sigma-Aldrich), anti-PhosphoSerine (9631s, Cell Signaling Technology), anti-pNLRP3$_{S198}$[12], anti-CSNK1A1 (301–991A, Bethyl Laboratories and H-7 sc-74582, Santa Cruz Biotechnology), HRP-anti-Mouse IgG (H + L) and HRP-anti-Rabbit IgG (H + L) (Promega), HRP-anti-Rat IgM+IgG (H + L) (Southern Biotech), and HRP-anti-Rabbit IgG(H + L) (Invitrogen). Data have been acquired on ChemiDoc Touch imaging system (BioRad) using Image Lab software (BioRad). Quantifications were performed with ImageJ. Uncropped and unprocessed scans of blots are available in the Source Data file. Antibodies are listed in Supplemental Table 4.

**Microscopy.** For ASC speck assays, cells were fixed with 4% paraformaldehyde (in PBS, 20 min), permeabilized in PBS, 0.1% Triton X-100, 0.2% Bovine serum albumin (BSA) (30 min), and stained with anti-ASC antibodies (2 μg/ml, Santa Cruz Biotechnology or 1 μg/ml, Adipogen) in PBS, 5% BSA (in PBS, 1 h) followed by anti-Rabbit IgG(H + L)-Alexa Fluor 660 or anti-Rabbit IgG(H + L)-Alexa Fluor 488 (6.7 μg/ml in PBS, 0.2% BSA, 30 min). Cells were imaged using AxioImager Z1 (Zeiss) and ×40 objectives. Images were quantified using ImageJ software. Caspase-1 activity was stained using FAM-FLICA Caspase-1 Assay kit (ImmunoChemisty Technologies) according to the manufacturer's instructions. Briefly, Hoechst 33342 (0.4 μg/ml, ImmunoChemistry Technologies) and FAM-FLICA reagent (1/50, 1 h) were added in the culture media upon inflammasome activation. Cells were washed once in DPBS (with calcium/ magnesium pH 7.1, Gibco), and media was replaced by DPBS/glycerol (1:1). Cells were imaged using CQ1 high content screening microscope (Yokogawa) and 10X objectives. Images were quantified using CQ1 and ImageJ softwares. Values are available in the Source Data file. Antibodies are listed in Supplemental Table 4.

**Cell death assay.** To assess cell permeabilization, cells were cultured in media without phenol red. PI (1.25 μg/ml, Immunochemistry technologies), YOPRO$^{TM}$-1 Iodide (1 μM, Invitrogen) and Hoechst (0.2 μg/ml, Immunochemistry technologies) were added 1 h before time-lapse imaging using CQ1 high content screening microscope (Yokogawa). Twi images/well were taken every 10 min using ×10 objectives. Images were quantified using the CQ1 software (Yokogawa). Pyroptosis was assessed using Cytotoxicity Detection Kit$^{PLUS}$ (LDH) (Roche) according to the manufacturer's instructions. Values are available in the Source Data file.

**ASC oligomerization assay.** BMDMs ($0.3 \times 10^6$ cells/ml) were resuspended in 20 mM HEPES pH 7.5, 10 mM KCl, 1.5 mM MgCl$_2$, 1 mM EDTA, 1 mM EGTA, 320 mM sucrose and EDTA-free protease inhibitor cocktail, and mechanically disrupted through 21 G needles 30 times. Lysates were centrifuged at 300×$g$ 8 min and supernatants were lysed in 1 volume of CHAPS buffer (20 mM HEPES, 5 mM MgCl$_2$, 0.5 mM EGTA, 0.1% CHAPS and EDTA-free protease inhibitor cocktail) and then centrifuged at 5000 × $g$ for 8 min. The pellets were washed once in PBS, resuspended in CHAPS buffer, crosslinked with disuccinimidyl suberate (DSS, 2 mM, 37 °C, 1 h) (Thermofisher), and then quenched in sample buffer 2X.

**Quantitative PCR.** BMDMs ($0.3 \times 10^6$ cells/ml) were lyzed in TRIzol (Invivogen) for total RNA purification. Single strand cDNAs were synthetized using ImProm-II™ Reverse Transcription System (Promega). QPCR were performed using iTaq™ Universal SYBR Green Supermix (Bio-Rad) on CFX96 Touch Real-Time PCR Detection System (Bio-Rad). Primers are listed in Supplementary Table 3. Data were analyzed with CFX Maestro software (Bio-Rad). Values are available in the Source Data file.

**Mice.** C57BL/6 J mice (Charles River laboratory) were housed at the PBES specific pathogen-free rodent facility (ENS Lyon) with food provided *ad libidum*, 13 h light/ 11 h dark cycle, 22 °C +/−2 °C, 50% +/−10% humidity.C57BL/6 J *Nlrp3*$^{−/−}$ mice were a kind gift from V. Petrilli (CRCL, Lyon). Experiments were performed in accordance with all national and institutional regulations for animal testing and research and protocols have been approved by the CECCAPP local ethic committee and the French *Ministère de l'Enseignement Supérieur, de la Recherche et de l'Innovation*. *Nlrp3*$^{S803D/S803D}$ KI mice were generated in-house by Crispr/CAS9 using UCUGAUUCCAAAGUCCCCCAGUUUUAGAGCUAUGCUGUUUUG (Eurogentec) as crRNA and TGTCCCATCAATGCTGCTTCGACATCTCCTCTGTCCTGAGCAGCAGCCAGAAGCTGGTGGAGCTGGACCTCgaTGACAATGCaCTGGGGGACTTTGGAATtcGATTGCTGTGTGTGGGACTGAAGCACCTGCTCTGCA as ssODN (Eurogentec) by electroporation of C57BL/6 J mouse blastocytes[41]. Mice were genotyped by specific PCR using TCACTGGCTGACTGAACGAC primer (primer R) with GGACCTCAGTGACAATGCC (primer WT-F) or TGGACCTCGATGACAATGCA (primer KI-F) primers. Genotyping were confirmed by unspecific PCR using primer R and ATCTCCCTAAGGTCACCCCC (primer F) followed by EcoR1 digestion. Mice were backcrossed for 5 generations with C57BL/6J. WT littermates were used as controls in all experiments. For endotoxic shock, 7-week old males *Nlrp3*$^{S803D/S803D}$, *Nlrp3*$^{−/−}$ and their respective *Nlrp3*$^{+/+}$ littermates were intraperitoneally injected with LPS (10 mg/kg). Mice survival was monitored each 12 h for 10 days and cytokine levels in the sera were measured at 2 and 4 h post-injection using Bio-Plex Pro Assays (Bio-Rad). *Brcc3*$^{−/−}$ mice were generated in-house by Crispr/CAS9 using AGUGAUGGGUCUGUGUAUAAGGUUUUAGAGCUAUGCUGUUUUG (Eurogentec) as crRNA by electroporation of C57BL/6 J mouse blastocytes, resulting in the frameshift deletion of 2 nucleotides in exon 1[41]. Mice were backcrossed for 7 generations with C57BL/6J, and genotyped by specific PCR using TCTGGCCTGGTTTGCAGTTT primer with GGAAGTGATGGGTCTGTGTATA (WT) or AAGTGATGGGTCTGTGTAGG (KO) primers.

**Structural models.** Models were based on a resolved structure and proposed models (PDB 6npy, with amino acid numbering changed for consistency, S804 corresponding to S806), and represented using PyMOL software[16]. The structures of NLRP3/NEK7 complex with different NLRP3 and NEK7 substitutions have been modeled using CHARMM-gui server (PDB reader)[42,43]. Electrostatic potential surfaces of each complex have been calculated using PBEQ solver from CHARMM-gui server. For the sake of clarity, the surface has been calculated for aa 742–864 of NLRP3 and 116–134 of NEK7. Molecular dynamics (MD) simulations were carried out on graphics processing units (GPUs) by means of high throughput MD (HTMD) software with CHARMM36m force field (trajectory of 100 ns). At the end of each simulation, the number of hydrogen bonds were extracted using Ligplot software.

**In vitro kinase assays.** GST-NLRP3 (1 μg, Abcam) was incubated of with GST-CSNK1A1 (100 ng, Sino Biological), GST-CSNK2A1 (100 ng, Sino Biological)/ GST-CSNK2B (100 ng, Creative Biomart), or 6-His-GST-CAMK2B (100 ng, Sino Biological)/Calmodulin (1 μg, Enzo) with 50 μM ATP and 4 μCi of [γ$^{32}$P]-ATP (PerkinElmer) in kinase buffer (50 mM HEPES pH 7.3, 100 mM NaCl, 10 mM MgCl$_2$, 0.05% Triton X-100, 10 mM β -glycerophosphate, 5 mM NaF) at 30 °C for 30 min in a final volume of 39 μL, supplemented with 2 mM CaCl$_2$ for CAMK2B/ Calmodulin reactions. Reactions were stopped by adding 15 μL of LDS (Thermo-Fisher Scientific) and 6 μL of DTT (Euromedex) 500 mM. Samples were analyzed by electrophoresis using Bolt 4–12% Bis-Tris Plus gel (ThermoFisher Scientific) followed by Coomassie blue gel staining using PageBlue™ Protein staining solution (ThermoFisher Scientific) and autoradiography.

**In cellulo phosphorylation assays.** In all, $5 \times 10^6$ U937 cells were treated with PMA (50 ng/ml) and doxycycline (2 μg/ml) for 16 h, followed by D4476 (20μM, 15 min before LPS), LPS (50 ng/ml, 4–5 h) and nigericin (15 μg/ml, 20 min). $1.2 \times 10^6$ 293 T cells were transfected with pcDNA3.1-HA-CSNK1A1 and pcDNA3-NLRP3 WT or S806A mutant for 24 h. Lysates (50 mM Tris-HCl pH 7.4,

150 mM NaCl, 2 mM EDTA, 1% NP-40, 1 mM PMSF, 1 mM NaF, 1 mM $Na_3VO_4$, EDTA-free protease inhibitor cocktail) were incubated with anti-NLRP3 and Protein A/G resin for 16 h. Beads were washed 6 times in lysis buffer and eluted in sample buffer 2X.

**Statistics**. Data were processed using Excel (Microsoft). ELISA and microscopy data were analyzed by ordinary two-way ANOVA with two-sided Tukey's multiple comparison test or repeated measure two-way ANOVA with two-sided Sidak's multiple comparison test (Prism v7). Cell deaths were analyzed with ordinary one-way ANOVA with two-sided Dunnett's multiple comparison test with single pooled variance. Survival curves were compared by two-sided Mantel-Cox test (Prism v7). Adjusted P values are listed in Supplementary Table 5.

**Reporting summary**. Further information on research design is available in the Nature Research Reporting Summary linked to this article.

## Data availability

All data supporting the findings of this study are provided in the Article and its Supplementary Information, or from the corresponding author on reasonable request. Source data are provided with this paper.

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

## Acknowledgements

This work was supported in part by the European Research Council (ERC-2013-CoG_616986 to B.F.P.), the Agence Nationale de la Recherche (ANR-13-JSV3-0002-01 to B.F.P., IDEXLYON of *Université de Lyon Investissements d'Avenir* program ANR-16-IDEX-0005 to B.F.P., AAPG 2017 LYSODIABETES to R.R., ANR-10-LABX-0030-INRT to R.R., ANR-10-IDEX-0002-02 to R.R., and ANR INGESTEM French National infra-structure to R.R.), the Finovi Foundation (to B.F.P.), the Chinese Scientific Council (201600090079 to T.N.), the European Foundation for the Study of Diabetes (EFSD)/Novo Nordisk Diabetes Research Programme grant (to R.R.) and by the "Fondation Pour La Recherche Médicale (FRM)" (EQU201903007859, Prix Roger Propice Prize, both to R.R.; DEQ20170336744 to V.P.). We thank Emad Alnemri (Thomas Jefferson University, Philadelphia, USA) for immortalized $Nlrp3^{-/-}$ and wild-type BMDMs, Jürg Tschopp and Fabio Martinon (UNIL, Lausanne, Switzerland) for the FLAG-NLRP3 constructs, Gabriel Nuñez (University of Michigan, Ann Arbor, USA) for immortalized $Nek7^{-/-}$ and wild-type BMDMs, and Tao Li (National Center for Biomedical Analysis, Beijing, China). We thank Isabelle Durieux (CIRI, Lyon, France) for help with subcloning. We thank Suzy Markossian and Marie Teixeira (SFR Biosciences, Lyon, France) for Crispr/CAS9-based mouse transgenesis. We acknowledge the contribution of SFR Biosciences (UMS3444/CNRS, US8/Inserm, ENS de Lyon, UCBL) PLATIM, PBES, genotyping and

flow cytometry facilities, as well as the Taplin mass spectrometry facility (Harvard Medical School, Boston).

## Author contributions

T.N., C.D.R., S.C., A.R., D.P., M.G., C.C., B.L., Z.Z., O.V. and S.H. performed the experimental studies and carried out the analysis. M.H. and O.W. performed the structural model studies. B.F.P., J.W., P.W., R.R., T.H., L.B. and V.P. supervised the work. B.F.P. and T.N. wrote the manuscript.

## Competing interests

The authors declare no competing interests.
