## [Peer Review File · Nature Communications]

NLRP3 phosphorylation in its LRR domain critically regulates inflammasome assemblyREVIEWER COMMENTS

Reviewer #1 (Remarks to the Author):

This is a very well conducted study. All appropriate controls are included giving confidence to the data and their interpretation. The authors convincingly identify that S803 phosphorylation and dephosphorylation is important for NLRP3 activation by recruiting NEK7 and subsequently the DUB BRCC3. The paper is well written and the data are clearly presented and articulated and the authors interpretations are supported by the data.

I have no major concerns regarding this work. My question to the authors is whether they tested, or whether they can speculate, on whether the S803/806 site is important for the NEK independent route described by a BioRxiv preprint (doi: <https://doi.org/10.1101/799320>). Some discussion of this should be included.

Reviewer #2 (Remarks to the Author):

In the manuscript "NLRP3 phosphorylation in its LRR domain critically regulates inflammasome Assembly", Tingting Niu and colleagues characterize S803 in mouse and S806 in human NLRP3 as a critical residue for recruiting essential cofactors for NLRP3 inflammasome assembly. Using mass spectrometry of the NLRP3 leucine-rich repeat (LRR) domain, 3 potential ubiquitination sites and 3 phosphorylation sites were identified. Using PMA differentiated U937 cells and immortalized BMDM with CRISPR/Cas9 deletion of NLRP3 as models, the authors restored cells with doxycycline-inducible WT NLRP3 or NLRP3 containing mutations of these identified residues. Mutation of S803, but not the other potential modified residues, revealed a defect NLRP3 response. Using a comprehensive and well controlled approach in mouse and human cells, the authors show that a phosphorylation mimicking mutation S803/806D or S803/806E prevents NLRP3 responses as determined by IL-1b and IL-18 secretion, caspase-1 activation and cleavage, pyroptosis and ASC polymerization, but not NLRC4 and AIM2 or inflammasome independent (TNF) responses. A S803/806A mutation showed only a partial defect, suggesting participation in a molecular switch. S803/806 is phosphorylated in a priming-dependent manner and de-phosphorylated upon NLRP3 activation. Using biochemical approaches, the authors define the mechanism by demonstrating that NLRP3 S803/806D is hyper-ubiquitinated and more efficiently degraded. Mechanistically, S803/806D fails to recruit the essential NEK7 in an activation-dependent manner, as this residue overlaps with the previously identified NEK7 binding site. Using the drug Oridonin, which prevents NEK7 recruitment, the authors then show that NEK7 recruitment is necessary to recruit the DUB BRCC3, which the authors identified earlier as a DUB for NLRP3 upon activation, which prevents degradation and enables NLRP3 responses. Based on sequence prediction, the authors screened a panel of 20 potential kinases by siRNA-mediated silencing and quantification of pyroptosis, which resulted in 4 potential hits: Csnk1a1, Csnk2b, Camk4 and Camk2b. From those, in vitro kinase assays demonstrated that Casein Kinase 1 alpha1 (CSNK1A1) phosphorylates NLRP3 on S806. The authors further generated a S803D mutant mouse by CRISPR/Cas9 targeting, which they show is completely resistant to LPS-induced shock phenocopying NLRP3 KO, which displays reduced IL-1b, IL-1a, IL-18, but not TNF or IL-6 release.

Hence, this manuscript provides evidence that S803/806 phosphorylation by CSNK1A1 during priming and then dephosphorylation during activation provide a molecular switch to enable NEK7 binding and subsequently binding of BRCC3 during activation to remove ubiquitin and consequently stabilize NLRP3 to facilitate NLRP3 inflammasome assembly. The complete defective NLRP3 response in the phosphorylation mimicking mutation and partial defect in the Ala mutation provides mechanistic insights into this switch, which has been shown earlier for other phosphorylated NLRP3 residues, such as S5 and S295, or Y859, the latter also preventing NEK7 interaction with NLRP3.

Overall, this study is performed very well, includes essential controls in human and mouse cells and provide novel insights into the mechanism stabilizing NLRP3. Below are several comments that would help to improve the conclusion of this manuscript.

1) A recent study in Nature Communications demonstrated that a so called human and mouse mini

NLRP3 (amino acids 1-686) lacking all LRRs is comparable in all canonical inflammasome responses, response kinetics as well as expression levels to full length NLRP3 (doi: 10.1038/s41467-018-07573-4). The response was equal with various priming and activation conditions. This would indicate that the LRRs should not contribute. Furthermore, interaction with NEK7 was intact and mediated by the NACHT and NLRP3 was also deubiquitinated at the NACHT. Oridonin also used in this study to support the sequential recruitment of NEK7 and BRCC3, modifies C279 within the NACHT (DOI: 10.1038/s41467-018-04947-6). In another study (doi.org/10.1038/s41467-019-11076-1), deletion of exon 5 by splicing (amino acids 714-776) renders NLRP3 defective, while earlier studies indicated that deletion of all LRRs renders NLRP3 active when ectopically expressed (DOI: 10.1016/j.cub.2004.10.027, DOI: 10.1074/jbc.M401178200). While it is feasible for the LRRs to provide a regulatory mechanism, additional support for these contradictory results would be required, as these studies are not easily reconcilable and contrasting WT, A803/806D and mini NLRP3 should be considered, which should provide additional insights and may help to clarify these contradictory results. The authors should also discuss this aspect in light of their new finding.

2) The title implies an inflammasome assembly regulation by LRR phosphorylation. While the downstream consequences of NLRP3 responses are analyzed, actual assembly of the core components is not. Several opportunities exist within presented data to investigate this aspect, where the authors purify NLRP3 (Fig. 2E, 5A-F, H, I, S5F, G) and could easily probe for ASC, which is inducible recruited to NLRP3.

3) The authors demonstrate that S803/806 is required for stability of NLRP3 and interaction with NEK7 and BRCC3, but provide limited evidence for NEK7-mediated BRCC3 binding to NLRP3. What is not completely clear is whether this defective interaction is a consequence of NLRP3 degradation and resulting loss of sensitivity to detect these interactions, as other DUBs are implicated in this response as well.

4) This aspect of sequential NEK7 and BRCC3 binding to NLRP3 is intriguing, but is insufficiently demonstrated. Support from over expression in HEK293 cells is limited. BRCC3 and NLRP3 are shown to interact in the absence of NEK7, but HEK293 cells also express NEK7 endogenously. To complement the Oridonin treatment, as it is not specific for NLRP3 and also inhibits NF- κ B and macrophage priming and could indirectly affect BRCC3 binding, NEK7 KO is necessary to investigate and support a conclusion of sequential recruitment and a dependence of BRCC3 on NEK7. Fig. 5H should also include analysis of NEK7 in the immunoprecipitated complex. In addition, Fig 5I could be improved with more equal immunoprecipitation of NLRP3 in samples with and without treatment of Oridonin by eventually analyzing an earlier time point of Nigericin treatment to minimize pyroptosis and NLRP3 release or treating cells with a caspase-1 inhibitor, such as VX-765.

5) In Fig 2E it is surprising that S806A is completely defective in Ser-phosphorylation, given that several other Ser residues were identified as being phosphorylated in a priming dependent manner in earlier studies. Do the authors imply that 803/806 is the primary residue responsible for subsequent phosphorylation on other S residues? This will require further investigation. In addition, less (about 50%) of S806A is immunoprecipitated compared to WT NLRP3, which may affect sensitivity of detection of phosphorylated NLRP3.

6) The S806A mutation only shows slightly reduced NEK7 binding (Fig 5B), but dramatically reduced IL-1b and IL-18 release. Furthermore, it does not result in any impact on ubiquitination, although it tremendously impairs BRCC3 binding. If, as proposed, these effects are connected and dependent on each other, one would expect a stronger correlation.

7) NLRP3 levels in WT and BRCC3 KO BMDM are comparable (Fig S5G), but one would expect a reduced expression in those cells, based on lack of BRCC3 binding and elevated ubiquitination.

8) Figure 7 demonstrates binding to- and in vitro phosphorylation of NLRP3 by CSNK1A1, but this aspect is underdeveloped compared to the rest of the manuscript. Missing are controls to confirm that CSNK1A1 does not affect transcriptional responses that would affect NLRP3 activation/responses and at least IL-1b, NLRP3, ASC, caspase-1 levels need to be shown, in

particular, as also TNF secretion is reduced by about 40% upon silencing of CSNK1A1. To compellingly demonstrate this functional connection, the authors should further investigate, whether priming-mediated NLRP3 S806 phosphorylation is defective following CSNK1A1 silencing, and demonstrate interaction of both proteins in U937 cells or BMDM, as one would expect this to be also priming dependent. Furthermore, silencing of CSNK1A1 should impair NEK7 and BRCC3 binding to NLRP3 comparable to the S803/806A mutant, and this aspect connecting all these findings is missing.

9) Figure 2C should include expression levels of WT, S806A, S806D and S806E.

10) Page 13, line 21: The statement that NEK7 was required for BRCC3 recruitment to NLRP3 is not supported by the referenced figure and the shown data: as shown it is enhanced, but not required.

11) HeLa cells are described in the methods section and should be removed, as no experiment is shown in HeLa cells.

Reply to the reviewers

Reviewer #1 (Remarks to the Author):

This is a very well conducted study. All appropriate controls are included giving confidence to the data and their interpretation. The authors convincingly identify that S803 phosphorylation and dephosphorylation is important for NLRP3 activation by recruiting NEK7 and subsequently the DUB BRCC3. The paper is well written and the data are clearly presented and articulated and the authors interpretations are supported by the data.

I have no major concerns regarding this work. My question to the authors is whether they tested, or whether they can speculate, on whether the S803/806 site is important for the NEK independent route described by a BioRxiv preprint (doi: <https://doi.org/10.1101/799320>). Some discussion of this should be included.

We thank the reviewer for the very positive assessment of our manuscript and his/her confidence in our data and their interpretations.

Schmacke et al. (doi: <https://doi.org/10.1101/799320>) describe a NEK-7 independent NLRP3 activation pathway in human cells (BlaER transdifferentiated macrophages, THP1 and iPS) and in immortalized mouse macrophages constitutively expressing NLRP3 in specific conditions of priming/activation signals. According to the authors' conclusion, NLRP3 priming may be pleiotropic, and a redundant TAK1-dependent priming pathway leading to NLRP3 PTM may bypass the requirement for NEK7. Given that the aforementioned work is not yet published, our request for NEK7-deficient BlaER1, THP1 and immortalized BMDMs constitutively expressing NLRP3 was declined, and we could not test whether NLRP3 S803 phosphorylation/dephosphorylation was relevant for the NEK7-independent inflammasome activation using the same cellular system as used in Schmacke et al.'s manuscript.

We used immortalized BMDMs from *Nek7^{-/-}* mice (kindly shared by G. Nunez, University of Michigan) to test whether IL-1 β secretion observed in our conditions was dependent on NEK7. NEK7^{-/-} BMDMs do not secrete IL-1 β in our conditions (LPS 50ng/ml 4h followed by Nigericin 15 μ g/ml 1h), while control immortalized BMDMs (also from G. Nunez's laboratory) secrete IL-1 β (Reply Fig 1). Noteworthy, control WT immortalized BMDMs do not secrete IL-1 β following LPS/Nig treatment conditions as described in Schmacke et al.'s study to trigger NEK7-independent NLRP3 activation in immortalized BMDMs constitutively expressing NLRP3 (LPS 200 ng/ml + Nig 7,25 μ g/ml (=10 μ M) added simultaneously for 4h). Therefore, constitutive NLRP3 expression may bypass the requirement of NEK7.

As S803 is located at the NLRP3/NEK7 interface and dephosphorylation at S803 is required for NLRP3/NEK7, we assume that NLRP3 S803 dephosphorylation might be mandatory only for NEK7-dependent activation. However, we cannot exclude that S803 site may be involved in interaction with other partner(s) or regulator(s) and may be critical for NEK7-independent inflammasome activation as well.

In the revised manuscript, we provide evidence that NLRP3 activation in our experimental settings depends on NEK7 in BMDMs (Supplementary Fig 5a) and discuss this point (page 19, line 403-409).

Reply Figure 1

Reply Fig. 1: Immortalized WT and *Nek7*^{-/-} BMDMs were either treated sequentially with LPS 50 ng/ml for 4h followed by nigericin 15 μ g/ml for 1h, or treated simultaneously with LPS 200 ng/ml and Nigericin 7.25 μ g/ml for 4h. IL-1 β and TNF secretions were measured by ELISA. NLRP3 and NEK7 expression were analyzed by WB. Means and 1 SD are represented, two-way ANOVA multiple comparisons of each condition with corresponding WT control, ****, $p < 0.0001$; n.d, non detectable.

Reviewer #2 (Remarks to the Author):

In the manuscript "NLRP3 phosphorylation in its LRR domain critically regulates inflammasome Assembly", Tingting Niu and colleagues characterize S803 in mouse and S806 in human NLRP3 as a critical residue for recruiting essential cofactors for NLRP3 inflammasome assembly. Using mass spectrometry of the NLRP3 leucine-rich repeat (LRR) domain, 3 potential ubiquitination sites and 3 phosphorylation sites were identified. Using PMA differentiated U937 cells and immortalized BMDM with CRISPR/Cas9 deletion of NLRP3 as models, the authors restored cells with doxycycline-inducible WT NLRP3 or NLRP3 containing mutations of these identified residues. Mutation of S803, but not the other potential modified residues, revealed a defect NLRP3 response. Using a comprehensive and well controlled approach in mouse and human cells, the authors show that a phosphorylation mimicking mutation S803/806D or S803/806E prevents NLRP3 responses as determined by IL-1 β and IL-18 secretion, caspase-1 activation and cleavage, pyroptosis and ASC polymerization, but not NLRC4 and AIM2 or inflammasome independent (TNF) responses. A S803/806A mutation showed only a partial defect, suggesting participation in a molecular switch. S803/806 is phosphorylated in a priming-dependent manner and de-phosphorylated upon NLRP3

activation. Using biochemical approaches, the authors define the mechanism by demonstrating that NLRP3 S803/806D is hyper-ubiquitinated and more efficiently degraded. Mechanistically, S803/806D fails to recruit the essential NEK7 in an activation-dependent manner, as this residue overlaps with the previously identified NEK7 binding site. Using the drug Oridonin, which prevents NEK7 recruitment, the authors then show that NEK7 recruitment is necessary to recruit the DUB BRCC3, which the authors identified earlier as a DUB for NLRP3 upon activation, which prevents degradation and enables NLRP3 responses. Based on sequence prediction, the authors screened a panel of 20 potential kinases by siRNA-mediated silencing and quantification of pyroptosis, which resulted in 4 potential hits: Csnk1a1, Csnk2b, Camk4 and Camk2b. From those, in vitro kinase assays demonstrated that Casein Kinase 1 alpha1 (CSNK1A1) phosphorylates NLRP3 on S806. The authors further generated a S803D mutant mouse by CRISPR/Cas9 targeting, which they show is completely resistant to LPS-induced shock phenocopying NLRP3 KO, which displays reduced IL-1b, IL-1a, IL-18, but not TNF or IL-6 release.

Hence, this manuscript provides evidence that S803/806 phosphorylation by CSNK1A1 during priming and then dephosphorylation during activation provide a molecular switch to enable NEK7 binding and subsequently binding of BRCC3 during activation to remove ubiquitin and consequently stabilize NLRP3 to facilitate NLRP3 inflammasome assembly. The complete defective NLRP3 response in the phosphorylation mimicking mutation and partial defect in the Ala mutation provides mechanistic insights into this switch, which has been shown earlier for other phosphorylated NLRP3 residues, such as S5 and S295, or Y859, the latter also preventing NEK7 interaction with NLRP3.

Overall, this study is performed very well, includes essential controls in human and mouse cells and provide novel insights into the mechanism stabilizing NLRP3. Below are several comments that would help to improve the conclusion of this manuscript.

We thank the reviewer for acknowledging the novelty of the insights into the mechanism of NLRP3 regulation provided by our work. We acknowledge him/her for highlighting the quality of our data. We greatly appreciate the reviewer's effort to comment our work and we believe that the additional work provided to address these comments strongly enforced the conclusion of our manuscript.

1) A recent study in Nature Communications demonstrated that a so called human and mouse mini NLRP3 (amino acids 1-686) lacking all LRRs is comparable in all canonical inflammasome responses, response kinetics as well as expression levels to full length NLRP3 (doi: 10.1038/s41467-018-07573-4). The response was equal with various priming and activation conditions. This would indicate that the LRRs should not contribute. Furthermore, interaction with NEK7 was intact and mediated by the NACHT and NLRP3 was also deubiquitinated at the NACHT. Oridonin also used in this study to support the sequential recruitment of NEK7 and BRCC3, modifies C279 within the NACHT (DOI: 10.1038/s41467-018-04947-6). In another study (doi.org/10.1038/s41467-019-11076-1), deletion of exon 5 by splicing (amino acids 714-776) renders NLRP3 defective, while earlier studies indicated that deletion of all LRRs renders NLRP3 active when ectopically expressed (DOI: 10.1016/j.cub.2004.10.027, DOI:

10.1074/jbc.M401178200). While it is feasible for the LRRs to provide a regulatory mechanism, additional support for these contradictory results would be required, as these studies are not easily reconcilable and contrasting WT, A803/806D and mini NLRP3 should be considered, which should provide additional insights and may help to clarify these contradictory results. The authors should also discuss this aspect in light of their new finding.

We thank the reviewer for giving us the opportunity to discuss the contradictory results of the literature concerning the role of the NLRP3 LRR domain. We acknowledge that all the aforementioned studies are indeed difficult to reconcile. In part, these discrepancies might be a consequence of the variety of the experimental systems : transient overexpression in 293T cells (Martinon et al., 2004, DOI: 10.1016/j.cub.2004.10.027), transient overexpression in THP1 cells (Dowds et al., 2004, DOI: 10.1074/jbc.M401178200), reconstitution of immortalized NLRP3-deficient BMDMs (Hafner-Bratkovic et al., 2018, DOI: 10.1038/s41467-018-07573-4), morpholino-treated primary human macrophages, reconstituted 293T and immortalized NLRP3-deficient BMDMs (Hoss et al., 2019, doi.org/10.1038/s41467-019-11076-1) and reconstitution of NLRP3-deficient U937 and immortalized NLRP3-deficient BMDMs and primary BMDMs from KI mice (our study). In addition, the various reconstitution methods and vectors of immortalized NLRP3-deficient BMDMs, may lead to differences in the levels of NLRP3 expression (Hafner-Bratkovic et al., 2018, DOI: 10.1038/s41467-018-07573-4 vs Hoss et al., 2019, doi.org/10.1038/s41467-019-11076-1 and our study). In order to compare NLRP3 S803/806A, NLRP3 S803/806D, mini-NLRP3 and NR-NLRP3 (non responsive NLRP3) in identical experimental system, we have cloned human and mouse mini-NLRP3 and NR-NLRP3 in pInducer21 and reconstituted NLRP3-deficient U937 and NLRP3-deficient immortalized BMDMs. In contrast to the published results (Hafner-Bratkovic et al., 2018, DOI: 10.1038/s41467-018-07573-4), mini-NLRP3 was not active in our two systems (Reply Fig 2a, b), supporting that NLRP3 LRR domain may be required for the inflammasome activation when NLRP3 is expressed at physiological level. Noteworthy, this result is consistent with Hoss et al., 2019 (doi.org/10.1038/s41467-019-11076-1).

After the released of all the aforementioned papers, a structural study has shown that NLRP3 interacts with NEK7 with multiple interfaces including the LRR, HD2 and NBD (Sharif et al, 2019, doi: 10.1038/s41586-019-1295-z). In this later study, additional co-IP experiments of mutant NEK7 with NLRP3 show that residues at the NEK7-NLRP3_{LRR} interfaces are critical for NEK7-NLRP3 binding and inflammasome activity, with substantial effect of residues at the NEK7-NLRP3_{HD2} interface while mutations of the residues at the NEK7-NLRP3_{NBD} interface have no effect on the interaction. No alternative mechanism of NLRP3 inflammasome assembly and NLRP3 regulation is provided in the study reporting dispensable NLRP3 LRR (Hafner-Bratkovic et al., 2018, DOI: 10.1038/s41467-018-07573-4). We could not exclude that mini-NLRP3 may be active when expressed at high level in reconstituted BMDMs (doi: 10.1038/s41467-018-07573-4) but LRR plays probably a critical role *in vivo* to stabilize NLRP3-NEK7 interaction. In our study, we generated NLRP3 S806D KI by crispr/CAS9 in mice, and used primary BMDMs, in order to avoid any artefacts linked to reconstitution and to maintain physiological *Nlrp3* gene regulation.

Oridonin has been shown to block the *in vitro* interaction between purified NLRP3 and NEK7 (DOI: 10.1038/s41467-018-04947-6). Oridonin targets NLRP3 C279 residue in the NACHT domain. Noteworthy C279 is not part of any of the interaction surfaces (*i.e.* LRR, HD2 or NBD interfaces)

between NEK7 and NLRP3 according to Sharif et al.'s structure (Reply Fig 2c). Therefore we may speculate that oridonin does not act by direct steric hindrance between NLRP3 NACHT and NEK7 but rather by inducing a conformational change disrupting NLRP3-NEK7 interaction. This conformational change could affect the NLRP3-NEK7 interfaces in the NACHT domain (HD2 or NBD interfaces) but also the interface in the LRR. Thus, the inhibitory effect of oridonin cannot be interpreted to support that the NACHT domain is the predominant interaction surface between NLRP3 and NEK7, which is not supported by structural data (Sharif et al, 2019, doi: 10.1038/s41586-019-1295-z).

In the discussion of the revised manuscript, we now mention the most recent publication supporting the dispensable role of LRR and provide technical hypothesis to the conflicting results (Hafner-Bratkovic et al., 2018, DOI: 10.1038/s41467-018-07573-4) (page 18, line 395-397).

Reply Fig. 2:

- NLRP3-deficient U937 cells reconstituted with doxycycline-inducible human NLRP3 WT, NLRP3 1-688 (mini-NLRP3), NLRP3 1-667 (NR-NLRP3), S806A or S806D mutant were treated with PMA (50 ng/ml) and doxycycline (2 μg/ml) for 16h, and then with LPS (50 ng/ml, 4h) followed by nigericin (15 μg/ml, 1h), or PMA (50 ng/ml) for 16h, and then doxycycline (2 μg/ml) and LPS (100 ng/ml) for 11h, before media change and addition of nigericin (7.25 μg/ml, 1h).

- b. Immortalized WT and *Nlrp3*^{-/-} BMDMs reconstituted with doxycycline-inducible mouse NLRP3 WT, NLRP3 1-686 (mini-NLRP3), NLRP3 1-665 (NR-NLRP3), S803A or S803D mutant were treated with either doxycycline (2 µg/ml) for 16h, and then with LPS (50 ng/ml, 4h) followed by nigericin (15 µg/ml, 1h), or with doxycycline (2 µg/ml) and LPS (100 ng/ml) for 11h, before media change and addition of nigericin (7.25 µg/ml, 1h). IL-1β and TNF secretions were measured by ELISA. NLRP3, pro-IL-1β, ASC and Caspase-1 expression were analyzed by WB. Means and 1 SD are represented, two-way ANOVA multiple comparisons of each condition with corresponding WT control, ****, p <0.0001.
- c. C279 (red) targeted by oridonin is not part of any of the interaction surfaces between NLRP3 (green) and NEK7 (blue). Structure model from PDB 6NPY (Sharif et al, 2019, doi: 10.1038/s41586-019-1295-z)

2) The title implies an inflammasome assembly regulation by LRR phosphorylation. While the downstream consequences of NLRP3 responses are analyzed, actual assembly of the core components is not. Several opportunities exist within presented data to investigate this aspect, where the authors purify NLRP3 (Fig. 2E, 5A-F, H, I, S5F, G) and could easily probe for ASC, which is inducible recruited to NLRP3.

By "inflammasome assembly" in the title we would like to refer to the full complex formation as analyzed by ASC specks observation and ASC oligomerization (Figs. 2d, 4e, f). We strengthen this point in the revised version of the manuscript. Indeed siRNA-mediated knock-down of CSNK1A1 expression strongly reduces the number of BMDMs with ASC speck following LPS+Nig treatment (Fig 7d), therefore inflammasome assembly depends on the CSNK1A1 kinase that phosphorylates NLRP3 at S803.

Interaction between the core components of the inflammasome in endogenous settings is technically challenging. Although the cause of these difficulties remains unclear, we can suspect both well-known detergent insolubility of oligomerized ASC filaments or transient interaction between NLRP3 and ASC oligomers. Therefore, our attempt to re-probe NLRP3 immunoprecipitations for ASC were unsuccessful (with samples from experiments displayed in initial Figs. 2E, 5A-F, H, I, S5F, G now Figs. 2e, 5a-f, i-j, supplementary Figs. 5g, j). In order to circumvent this challenge, and to assess NLRP3 S803D ability to directly recruit ASC, we provide in the revised version new co-IP experiments using hypotonic buffer (Figure 4g). As expected, the results confirm that NLRP3 S803D does not recruit ASC in primary BMDMs following LPS+Nig.

3) The authors demonstrate that S803/806 is required for stability of NLRP3 and interaction with NEK7 and BRCC3, but provide limited evidence for NEK7-mediated BRCC3 binding to NLRP3. What is not completely clear is whether this defective interaction is a consequence of NLRP3 degradation and resulting loss of sensitivity to detect these interactions, as other DUBs are implicated in this response as well.

In the experimental settings used to assess NLRP3/NEK7 binding in primary BMDMs (Fig 5a, lanes 4 and 6) or U937 (Fig 5b, lanes 3 and 7), similar level of NLRP3 WT and NLRP3 S803D is detected in the cell lysates. Similarly, in the experimental settings used to assess NLRP3/BRCC3 binding in primary BMDMs (Fig 5c, lanes 3 and 6) or U937 (Fig 5e, lanes 4 and 8), level of NLRP3 S806D in

the cell lysates is only slightly decreased as compared to NLRP3 WT. So, it is unlikely that the total impairment of the co-IPs between NLRP3 S803D-NEK7 and NLRP3 S803D-BRCC3 are the consequences of its decrease stability. In the revised manuscript, we provide quantification of the amount of NLRP3 in the IPs normalized to the amount of NLRP3 in the lysates.

In order to provide additional evidence on NEK7-mediated BRCC3 binding to NLRP3, we performed BRCC3-NLRP3 co-IP in *Nek7*^{-/-} immortalized BMDMs (Fig. 5h). NLRP3 does not co-immunoprecipitate with BRCC3 in *Nek7*^{-/-} immortalized BMDMs, confirming that NLRP3-BRCC3 interaction depends on NEK7. We thank the reviewer for giving us the opportunity to clearly demonstrate this important novel insight into NLRP3 regulation mechanism described in our study.

4) This aspect of sequential NEK7 and BRCC3 binding to NLRP3 is intriguing, but is insufficiently demonstrated. Support from over expression in HEK293 cells is limited. BRCC3 and NLRP3 are shown to interact in the absence of NEK7, but HEK293 cells also express NEK7 endogenously. To complement the Oridonin treatment, as it is not specific for NLRP3 and also inhibits NF- κ B and macrophage priming and could indirectly affect BRCC3 binding, NEK7 KO is necessary to investigate and support a conclusion of sequential recruitment and a dependence of BRCC3 on NEK7. Fig. 5H should also include analysis of NEK7 in the immunoprecipitated complex. In addition, Fig 5I could be improved with more equal immunoprecipitation of NLRP3 in samples with and without treatment of Oridonin by eventually analyzing an earlier time point of Nigericin treatment to minimize pyroptosis and NLRP3 release or treating cells with a caspase-1 inhibitor, such as VX-765.

As requested by the reviewer, we performed BRCC3-NLRP3 co-IP in *Nek7*^{-/-} immortalized BMDMs (Fig. 5h). This new experiment shows that NLRP3 does not co-immunoprecipitate with BRCC3 in *Nek7*^{-/-} immortalized BMDMs, confirming our previous conclusion based on co-overexpression and oridonin treatment that NLRP3-BRCC3 interaction depends on NEK7. We thank the reviewer by pointing out this key experiment that indeed was necessary to prove the sequential recruitment of NEK7 and BRCC3.

In addition, we analyzed the presence of NEK7 in the anti-BRCC3 precipitation (initial Fig 5h, now Fig 5i). Despite intensive efforts using 3 different anti-NEK7, we could not detect NEK7 in the immunoprecipitated, probably in part because NEK7 band is hidden by non-specific background bands detectable in NEK7 KO IP and IgG control IP (reply Fig 3). In addition, as all these IPs were performed on endogenous proteins to avoid artefact of overexpression, the amount of co-immunoprecipitated proteins are quite low and hard to detect, especially indirect interaction in tripartite complex. Although we cannot fully conclude that NLRP3/NEK7/BRCC3 form a tripartite complex or successive complexes, our data strongly demonstrate the sequential recruitment of NEK7 and BRCC3. This point is now discussed in the revised discussion (page 19, line 415-416).

Following the reviewer's advice, we repeated the experiment displayed in Fig. 5j (initial Fig. 5i) with addition of caspase-1 inhibitor VX-765 before Nig treatment in order to minimize pyroptosis and manage to achieve equal immunoprecipitations with/without oridonin. In addition, we now provide a quantification of the Ub intensity normalized to the NLRP3 intensity in the IP. This experiment fully confirms that oridonin increases NLRP3 ubiquitination. We thank the reviewer for his/her precious advice.

Reply Fig.3: Immortalized WT and *Nek7*^{-/-} BMDMs were primed with LPS (50 ng/ml, 6h) and treated with nigericin (15 µg/ml, 30 min). Endogenous BRCC3 immunoprecipitates were analyzed for NLRP3 by WB. Lysate of immortalized WT BMDMs incubated with isotype control and protein A-beads (iBMDMs WT+IgG), and BRCC3 immunoprecipitates from lysate of immortalized *Nlrp3*^{-/-} BMDMs were used as a negative control. In BRCC3 immunoprecipitates, background bands prevented the detection of NEK7 specific band.

5) In Fig 2E it is surprising that S806A is completely defective in Ser-phosphorylation, given that several other Ser residues were identified as being phosphorylated in a priming dependent manner in earlier studies. Do the authors imply that 803/806 is the primary residue responsible for subsequent phosphorylation on other S residues? This will require further investigation. In addition, less (about 50%) of S806A is immunoprecipitated compared to WT NLRP3, which may affect sensitivity of detection of phosphorylated NLRP3.

In the initial Fig 2e, unequal ECL background of the IP:NLRP3 WB:NLRP3 artificially increases the intensity of the NLRP3 signals in lanes 3-4 as compared to lanes 6-7, while NLRP3 signal in the total lysate is equal in lanes 3 and 6. To solve this issue, we reloaded and reblotted IP:NLRP3 samples with anti-NLRP3, and the new WB shows that equal amounts of WT and S806A mutant are immunoprecipitated in lanes 3 and 6 (in accordance with equal amount of NLRP3 in the lysates lane 3 and 6). In addition, we now quantify IP:NLRP3 WB:P-Ser normalized to IP:NLRP3 WB:NLRP3, and confirm that S806A is indeed completely defective in Ser-phosphorylation.

In order to independently confirm this result, we additionally tested the impact of pharmacological inhibition of CSNK1A1 by D4476 on NLRP3 Serine phosphorylation (Fig. 7i). Consistent with NLRP3 S806A lack of phosphorylation, D4476 treatment prior LPS priming completely abolishes p-Ser NLRP3 signal in LPS-primed U937. Therefore, our results suggest that either S806/3 is the only NLRP3 phosphorylated serine following priming, or phosphorylation of NLRP3 S806/3 by CSNK1A1 is a prerequisite for subsequent phosphorylation on other S residues. This point is now addressed in the revised discussion (page 18, line 376-380).

As suggested by the reviewer, we have tried to investigate this question further. To the best of our knowledge, only one Ser (Ser198 (human) corresponding to Ser194 (mouse)) in NLRP3 has been reported to be phosphorylated in a priming dependent manner. NLRP3 Ser198/4 is phosphorylated by JNK1 (Song et al., 2017.doi: 10.1016/j.molcel.2017.08.017). In order to test the phosphorylation of mutant NLRP3 S806/3D on S198/4 upon LPS priming in our experimental settings, we used polyclonal phospho-specific human pNLRP3_{S198} antibodies kindly shared by Tao Li (Beijing, China). We were not able to detect pNLRP3_{S198} in conditions that prime U937 in our

study (PMA (50 ng/ml, 16h)-differentiated U937 cells treated with LPS (50 ng/ml, 6h)) (reply Fig 4a). As a positive control to validate the phospho-specific human pNLRP3_{S198} antibodies, pNLRP3_{S198} can be detected in 293T ectopically expressing NLRP3 (reply Fig 4b). Song et al. reported NLRP3 phosphorylation at S198 in U937 in different priming conditions (*i.e* PMA 154 ng/ml, 24h in the absence of LPS). These different experimental conditions could account for the differences in the S198 phosphorylation status. The polyclonal phospho-specific mouse pNLRP3_{S194} antibodies generated by the same lab is no longer available to be shared and therefore we were unable to directly test the phosphorylation of NLRP3 at S194 in BMDMs. As an alternative approach to investigate the phosphorylation of NLRP3 at S194 in BMDMs using our experimental settings, we tested the impact of pharmacological inhibition and knock-down of the JNK1 on IL-1 β secretion by LPS+Nigericin treated BMDMs. JNK1 inhibitor SP600125 did not reduce the secretion of IL-1 β by LPS+Nigericin-treated BMDMs even when SP600125 was used in 2.5X excess concentration than in Song et al. (40nM). Consistently, knock-down of JNK1 expression (encoded by *Mapk8*) also showed no effect in reducing IL-1 β secretion by LPS+Nigericin-treated BMDMs. Although, these results are difficult to reconcile with Song et al.'s results, they are consistent with the complete inhibition of P-Ser NLRP3 in NLRP3 S806A expressing U937. Generation of new batch of phospho-specific mouse pNLRP3_{S194} would be required to further investigate this point.

Reply Fig. 4:

a. NLRP3-deficient U937 cells reconstituted with doxycycline-inducible NLRP3 were treated with PMA (50 ng/ml) and doxycycline (2 μ g/ml) for 16h, and then with LPS (50 ng/ml, 6h). Anti-NLRP3 immunoprecipitates were analyzed for pNLRP3_{S198} using phospho-specific human pNLRP3_{S198} antibodies by WB.

b. 293T cells ectopically expressing human NLRP3 WT or S806D mutant were analyzed by anti-NLRP3 immunoprecipitation followed by anti-pNLRP3_{S198} WB.

c. BMDMs were treated with SP600125 15 min prior priming with LPS (50 ng/ml, 4h) followed by nigericin (15 μ g/ml, 1h). IL-1 β and TNF secretions were measured by ELISA.

d. BMDMs transfected with indicated siRNA and treated with LPS (50 ng/ml, 4h) followed by

nigericin (15 μ g/ml, 1h). IL-1 β and TNF secretions were measured by ELISA. Means and 1 SD are

represented, two-way ANOVA multiple comparisons of each condition with corresponding non targeting (NT) siRNA control, ****, $p < 0.0001$.

6) The S806A mutation only shows slightly reduced NEK7 binding (Fig 5B), but dramatically reduced IL-1 β and IL-18 release. Furthermore, it does not result in any impact on ubiquitination, although it tremendously impairs BRCC3 binding. If, as proposed, these effects are connected and dependent on each other, one would expect a stronger correlation.

We sincerely thank the reviewer for pointing out our conclusion and model concerning the impairment of S806A mutant was unclear in the initial version of the manuscript, and giving us the opportunity to provide additional insights on this point.

While both NLRP3 S806A and S806D mutations abolish IL-1 β and IL-18 secretion, the underneath mechanism is obviously distinct. Our results show that NLRP3 S806D mutation impairs NLRP3/NEK7 interaction by modification of their binding interface, resulting in the inability for NLRP3 S806D to recruit BRCC3 leading to NLRP3 S806D ultimate degradation. On the opposite NLRP3 S806A mutation does not impair the interaction with NEK7 (Fig 5b, lane 5 and supplementary Fig 5c), consistently with the structural data of the NLRP3-NEK7 interaction surface (Supplementary Fig 5b, e, f). Significant amount of NLRP3 S806A co-immunoprecipitates with BRCC3 (Fig 5e, lane 6), resulting consistently in very moderate increase in its ubiquitination following LPS+Nig treatment (Fig 5 f, lane 8-9). Therefore, the following sequence of events "NEK7 recruitment-BRCC3 recruitment-Deubiquitination" seems functional for NLRP3 S806A mutant, and NLRP3 S806A is inactive despite its ability to recruit NEK7. Our interpretation of these data is that NLRP3 phosphorylation at S806 by CSNK1A1 participates in the priming that makes NLRP3 competent for activation, while NLRP3 dephosphorylation at S806 is required for NLRP3 to bind to NEK7 upon activation. This is consistent with the transient phosphorylation of NLRP3 upon LPS priming and its dephosphorylation upon activation (Fig. 2e). As aforementioned, NLRP3 S806A mutation may completely inhibit subsequent phosphorylation on other S residues following LPS priming. We can make the hypothesis that some of these post-translational modifications is key for NLRP3 priming.

Noteworthy, two other phosphorylation sites (S3/5 and S291/295) in NLRP3 have been shown to be inactive both in their phospho-null and phospho-mimetic forms by distinct mechanisms. (1) Both NLRP3 S3A and NLRP3 S3D or S3E are inactive. Phospho-mimetic NLRP3 S3D/S3E mutations modify the electrostatic properties of the NLRP3/ASC interaction surface, rendering NLRP3 S3D defective in recruiting ASC (Stutz et al., 2017 doi: 10.1084/jem.20160933). Phospho-null NLRP3 S3A mutation impairs S3 phosphorylation that is protective against TRIM31-mediated NLRP3 degradation (Zhao et al., 2020 doi: 10.4049/jimmunol.2000649). (2) Phosphorylation of NLRP3 at S291 regulates NLRP3 transient recruitment with the membrane compartment and its later release to the cytosol, and both NLRP3 S291A and NLRP3 S291E are inactive (Zhang et al., 2017 doi: 10.1084/jem.20162040). NLRP3 S291E mutant fails to be recruited to the MAMs and remains diffuse in the cytosol, while NLRP3 S291A fails to be released from the Golgi and to form cytosolic pyroptosome.

We thank the reviewer for pointing out this part of our results may require additional discussion now provided in the revised manuscript (page 17, line 372-373 and page 18, line 380-384).

7) NLRP3 levels in WT and BRCC3 KO BMDM are comparable (Fig S5G), but one would expect a reduced expression in those cells, based on lack of BRCC3 binding and elevated ubiquitination.

We thank the reviewer for raising this point that ultimately helped us to reconcile our current and past findings.

As expected, levels of **WT** NLRP3 are indeed similar in *Brcc3*^{-/-} and *Brcc3*^{+/+} BMDMs (supplementary Fig 5j, initial FigS5G). We confirmed this observation in the presence of caspase-1 inhibitor VX-765 to circumvent artefacts caused by pyroptosis in *Brcc3*^{+/+} BMDMs (supplementary Fig 5h). In addition, NLRP3 level does not increase in *Brcc3*^{-/-} BMDMs following treatment with inhibitors of ubiquitin-dependent degradation pathways (MG132 and E-64d). These results are entirely consistent with the conclusion of our previous publication (Py et al., 2013 doi: 10.1016/j.molcel.2012.11.009) that NLRP3 ubiquitination controlled by BRCC3 is inhibitory without targeting **WT** NLRP3 to degradation. Indeed, MG132 and/or E64d does not increase the amount of ubiquitinated NLRP3 when BRCC3 is inhibited by G5 (Py et al., 2013 doi: 10.1016/j.molcel.2012.11.009).

In contrast, results presented in this manuscript support that the defect of NLRP3 **S806D** to recruit BRCC3 leads to degradative ubiquitination as ubiquitinated NLRP3 **S806D** accumulates upon treatment with MG132 and E64d. Thus, blockade of BRCC3-mediated deubiquitination leads to different outcomes for NLRP3 **WT** and **S806D** mutant, probably because they associate differently with partner NEK7, therefore form different complexes and are decorated by different ubiquitin chains. To check this hypothesis, we compared the type of ubiquitin chains associated with NLRP3 **WT** and **S806D** mutant upon LPS+Nig treatment and the BRCC3 inhibitor G5. Interestingly indeed, NLRP3 S806D is mainly associated with K48 ubiquitin chains in *Nlrp3*^{S806D/S806D} BMDMs treated with LPS+Nig, while NLRP3 WT is mainly associated with K63 chains in *Nlrp3*^{+/+}BMDMs treated with LPS+Nig+G5 (Supplementary Fig5i). These new results, included (Supplementary Figs. 5h, i) and discussed (page 14, line 282-291) in the revised manuscript, reconcile our past and current findings.

8) Figure 7 demonstrates binding to- and in vitro phosphorylation of NLRP3 by CSNK1A1, but this aspect is underdeveloped compared to the rest of the manuscript. Missing are controls to confirm that CSNK1A1 does not affect transcriptional responses that would affect NLRP3 activation/responses and at least IL-1b, NLRP3, ASC, caspase-1 levels need to be shown, in particular, as also TNF secretion is reduced by about 40% upon silencing of CSNK1A1. To compellingly demonstrate this functional connection, the authors should further investigate, whether priming-mediated NLRP3 S806 phosphorylation is defective following CSNK1A1 silencing, and demonstrate interaction of both proteins in U937 cells or BMDM, as one would expect this to be also priming dependent. Furthermore, silencing of CSNK1A1 should impair NEK7 and BRCC3 binding to NLRP3 comparable to the S803/806A mutant, and this aspect connecting all these findings is missing.

We agree with the reviewer that this part was underdeveloped in the initial version of the manuscript and we provide now additional data in the revised version. As asked by the reviewer we assessed the transcriptional response upon Csnk1a1 knock-down downstream of LPS by

following the induction of *Il-1b*, *Nlrp3*, *Asc*, *Casp1* and *Tnf* levels by qPCR in BMDMs (Supplementary Fig. 7b). These controls show that knock-down of *Csnk1a1* does not reduce the induction of *Il-1b*, *Nlrp3*, *Casp1*, or affects *Asc* expression. In addition, *Csnk1a1* knock-down does not decrease the induction of *Tnf* downstream LPS. Noteworthy, the 40% reduction of TNF secretion as initially observed during the siRNA screen (Fig 7b) was most likely caused by a technical artefact as (1) during the screen it was observed with all the different targeting siRNAs including the *Nlrp3* siRNA control and not specific to si*Csnk1A1* (Fig 7b) and (2) this was not reproducible upon following up experiments (Fig 7c). Concerning the connection between CSNK1A1 and NLRP3 phosphorylation upon priming, we could not address this point using *Csnk1a1* knock-down given the high cost of siRNA to transfect the large amount of cells required to visualize NLRP3 phosphorylation (minimum 5×10^6 cells/condition). Alternatively, we choose a pharmacological approach and show that indeed pharmacological CSNK1A1 inhibitor D4476 completely inhibited priming-mediated NLRP3 S806 phosphorylation (Fig. 7i). Finally, we investigated the interaction between endogenous NLRP3 and CSNK1A1 by co-immunoprecipitation in LPS-BMDMs. As expected, NLRP3 co-immunoprecipitated with CSNK1A1 in BMDMs in a LPS priming-dependent manner (Fig. 7g). As explained in our reply to question#6, NLRP3 S806A mutant co-immunoprecipitates with NEK7 and BRCC3. Therefore, we do not expect that silencing CSNK1A1 would alter NLRP3 co-immunoprecipitation with both partners. We hope that the explanation and the model as proposed in the reply to question#6 would satisfy the reviewer.

In addition to the points specifically asked by the reviewer, we additionally strengthened the NLRP3/CSNK1 aspect in the revised version by using alternative readouts of inflammasome activation. Our new results show that knock-down of *Csnk1a1* reduces NLRP3-dependent IL-18 secretion as well as ASC speck formation in BMDMs treated with LPS and nigericin (Figs. 7c, d).

9) Figure 2C should include expression levels of WT, S806A, S806D and S806E.

We thank the reviewer for pointing out this missing controls. Lysates of this experiment have been re-blotted for anti-NLRP3 and this control is now provided in the revised manuscript (Fig. 2c).

10) Page 13, line 21: The statement that NEK7 was required for BRCC3 recruitment to NLRP3 is not supported by the referenced figure and the shown data: as shown it is enhanced, but not required.

We agree with the reviewer that the previous claim was overstated. As discussed in question#3 and #4, we now provided evidence that NEK7 is required for BRCC3 recruitment to NLRP3 by using NEK7-deficient immortalized BMDMs (Fig. 5h). The text of the manuscript was revised accordingly.

11) HeLa cells are described in the methods section and should be removed, as no experiment is shown in HeLa cells.

We are thankful to the reviewer for noticing this discrepancy. Co-immunoprecipitations between ectopically expressed NLRP3 and endogenous NEK7 have been done in HeLa cells and not in 293T cells has mentioned by mistake in the initial version of our manuscript. We sincerely apologize for this error. We changed the main text, methods and figure legend accordingly in the revised manuscript. In addition, during the reviewing process we independently repeat this experiment to present the results without having to "cut" the image of the membrane. The new version of the

supplementary figure 5c show identical results with all conditions side by side in one image. We hope that the reviewer will appreciate our effort to improve the quality of our manuscript.

REVIEWERS' COMMENTS

Reviewer #2 (Remarks to the Author):

Niu and colleagues provide a very extensive revision addressing all concerns raised by both reviewers with additional high quality experiments and extensive discussion. I do not have any further concerns and this study is an important contribution to the field.

Minor:

The authors performed important experiments to reconcile some of the published conflicting results and provided these data for this review purpose. I believe some of those data should be included as supplementary figures, as this is very informative and helps to explain some of their observations.

1) Reply figure 2 supports the essential role of the NLRP3 LRR, as demonstrated by an appropriate inducible expression system matching endogenous NLRP3 expression, which I believe is important for the premise of the manuscript.

2) Reply figure 4 supports their finding of S806/3 being the dominant (only?) relevant Ser phosphorylation site by extensively analyzing earlier reported S198/4 by using p-specific antibodies, pharmacological JNK1 inhibition and siRNA silencing of MAPK8 and should be included in the manuscript.

RESPONSE TO THE REVIEWERS' COMMENTS

Reviewer #2 (Remarks to the Author):

Niu and colleagues provide a very extensive revision addressing all concerns raised by both reviewers with additional high quality experiments and extensive discussion. I do not have any further concerns and this study is an important contribution to the field.

We thank the reviewer for acknowledging the quality of our revision.

Minor:

The authors performed important experiments to reconcile some of the published conflicting results and provided these data for this review purpose. I believe some of those data should be included as supplementary figures, as this is very informative and helps to explain some of their observations.

1) Reply figure 2 supports the essential role of the NLRP3 LRR, as demonstrated by an appropriate inducible expression system matching endogenous NLRP3 expression, which I believe is important for the premise of the manuscript.

As suggested by the reviewer, we have included these data in the revised version of the manuscript as Supplementary Fig. 2b (previously Reply Fig. 2a) and Supplementary Fig. 3b (previously Reply Fig. 2b).

2) Reply figure 4 supports their finding of S806/3 being the dominant (only?) relevant Ser phosphorylation site by extensively analyzing earlier reported S198/4 by using p-specific antibodies, pharmacological JNK1 inhibition and siRNA silencing of MAPK8 and should be included in the manuscript.

As suggested by the reviewer, we have included these data in the revised version of the manuscript as Supplementary Figs. 2 h, i (previously Reply Fig. 4a-b), and Supplementary Figs. 7a, b (previously Reply Fig. 4c-d).